# The 2019 Raikoke volcanic eruption: Part 1 Dispersion model simulations and satellite retrievals of volcanic sulfur dioxide

Johannes de Leeuw[1], Anja Schmidt[1,2], Claire S. Witham[3], Nicolas Theys[4], Isabelle A. Taylor[5], Roy G. Grainger[5], Richard J. Pope[6,7], Jim Haywood[3,8], Martin Osborne[3,8], and Nina I. Kristiansen[3]

[1]Department of Chemistry, University of Cambridge, Cambridge, UK.
[2]Department of Geography, University of Cambridge, Cambridge, UK.
[3]Met Office, Exeter, UK
[4]Royal Belgian Institute for Space Aeronomy (BIRA-IASB), Brussels, Belgium.
[5]COMET, Sub-Department of Atmospheric, Oceanic and Planetary Physics, University of Oxford, Oxford, UK
[6]School of Earth and Environment, University of Leeds, Leeds, UK.
[7]National Centre for Earth Observation, University of Leeds, Leeds, UK.
[8]College of Engineering, Mathematics, and Physical Sciences, University of Exeter, Exeter, UK.

**Correspondence:** Johannes de Leeuw (jd876@cam.ac.uk)

**Abstract.** Volcanic eruptions can cause significant disruption to society and numerical models are crucial for forecasting the dispersion of erupted material. Here we assess the skill and limitations of the Met Office's Numerical Atmospheric-dispersion Modelling Environment (NAME) in simulating the dispersion of the sulfur dioxide ($SO_2$) cloud from the 21-22 June 2019 eruption of the Raikoke volcano (48.3°N, 153.2°E). The eruption emitted around $1.5 \pm 0.2$ Tg of $SO_2$, which represents the largest volcanic emission of $SO_2$ into the stratosphere since the 2011 Nabro eruption. We simulate the temporal evolution of the volcanic $SO_2$ cloud across the Northern Hemisphere (NH) and compare our model simulations to high-resolution $SO_2$ measurements from the Tropospheric Monitoring Instrument (TROPOMI) and the Infrared Atmospheric Sounding Interferometer (IASI) satellite $SO_2$ products.

We show that NAME accurately simulates the observed location and horizontal extent of the $SO_2$ cloud during the first 2-3 weeks after the eruption, but is unable, in its standard configuration, to capture the extent and precise location of the highest magnitude vertical column density (VCD) regions within the observed volcanic cloud. Using the Structure-Amplitude-Location (SAL) score and the Fractional Skill Score (FSS) as metrics for model skill, NAME shows skill in simulating the horizontal extent of the cloud for 12-17 days after the eruption where VCDs of $SO_2$ (in Dobson Units, DU) are above 1 DU. For $SO_2$ VCDs above 20 DU, which are predominantly observed as small-scale features within the $SO_2$ cloud, the model shows skill on the order of 2-4 days only. The lower skill for these high $SO_2$ VCD regions is partly explained by the model-simulated $SO_2$ cloud in NAME being too diffuse compared to TROPOMI retrievals. Reducing the standard horizontal diffusion parameters used in NAME by a factor of four results in a slightly increased model skill during the first five days of the simulation, but on longer timescales the simulated $SO_2$ cloud remains too diffuse when compared to TROPOMI measurements.

The skill of NAME to simulate high $SO_2$ VCDs and the temporal evolution of the NH-mean $SO_2$ mass burden is dominated by the fraction of $SO_2$ mass emitted into the lower stratosphere, which is uncertain for the 2019 Raikoke eruption. When emitting 0.9-1.1 Tg of $SO_2$ into the lower stratosphere (11-18 km) and 0.4-0.7 Tg into the upper troposphere (8-11 km), the

NAME simulations show a similar peak in $SO_2$ mass burden to that derived from TROPOMI (1.4-1.6 Tg of $SO_2$) with an average $SO_2$ e-folding time of 14-15 days in the NH.

Our work illustrates how the synergy between high-resolution satellite retrievals and dispersion models can identify potential
limitations of dispersion models like NAME, which will ultimately help to improve dispersion modelling efforts of volcanic $SO_2$ clouds.

## 1 Introduction

Volcanic activity can vary strongly in intensity, ranging from passive degassing volcanoes emitting sulfur into the lower troposphere to explosive eruptions that can release large amounts of ash and gases high into the stratosphere (e.g., Oppenheimer
et al., 2011). It is well established that volcanic eruptions can impact Earth's climate system through changes in the energy balance (e.g., Robock, 2000; Schmidt et al., 2012; Stenchikov, 2016; Schmidt et al., 2018), which can affect the hydrological cycle (e.g., Trenberth and Dai, 2007) and atmospheric dynamics (e.g., Shindell et al., 2004). Furthermore, volcanic air pollution events can lead to a severe and spatially widespread health hazard and increase excess mortality (e.g., Schmidt et al., 2011).

For the aviation industry, ash and gas emissions from volcanic eruptions can pose a flight safety hazard. Flying through
volcanic ash is a well-recognised hazard as it can reduce visibility, damage the exterior of the aircraft and compromise the functionality of aircraft engines. Ingestion of volcanic ash can cause engine failure and permanently damage jet engines (Casadevall et al., 1996; Prata and Tupper, 2009; Dunn, 2012; Prata et al., 2019). When sulfur dioxide ($SO_2$) oxidises to sulfuric acid and upon hydration forms sulfuric acid aerosol particles (see e.g., Hamill et al., 1977; Hofmann and Rosen, 1983), damage to the exterior of aircraft can occur (e.g. window crazing) (Bernard and Rose, 1990). Through sulfidation, $SO_2$ can also cause serious
damage to the interior of the engines. Sulfuric acid aerosol particles have been recorded to corrode nickel alloys in engine components (e.g. compressor blades) when alkali metal salts, like mineral dust or sea salt, are co-present (Eliaz et al., 2002; Grégoire et al., 2018). While this effect has not been linked to immediate engine failures, it is a concern for the aviation industry as it increases maintenance costs. Apart from material damage, sulfurous odours can also cause distress of cabin passengers and aircrew.

The Raikoke volcano is located in the Kuril Island chain, near the Kamchatka Peninsula in Russia (48.3°N, 153.2°E, see fig. 1) and had been dormant since 1924. On June 21, 2019 at 1805 UTC Raikoke started erupting and multiple explosions were reported until 0540 UTC on 22 June, 2019 (Crafford and Venzke, 2019; Hedelt et al., 2019). During this period, Raikoke released the largest amount of $SO_2$ into the stratosphere since the Nabro eruption in 2011 (Goitom et al., 2015). The volcanic cloud (while a geographic distribution of $SO_2$ and/or sulfate aerosol is not technically a cloud in the meteorological sense,
we use this term through our paper as it is common practice in the atmospheric dispersion community) dispersed across the Northern Hemisphere (NH) within the first few weeks after the eruption and was observed by various ground-based observational networks (e.g., Vaughan et al., 2020; Mateshvili et al., 2020), aircraft-based instruments (e.g., Bundke et al., 2020) and satellites (e.g., Muser et al., 2020; Hyman and Pavolonis, 2020; Prata et al., 2020; Kloss et al., 2021; Gorkavyi et al., 2021) in the following months.

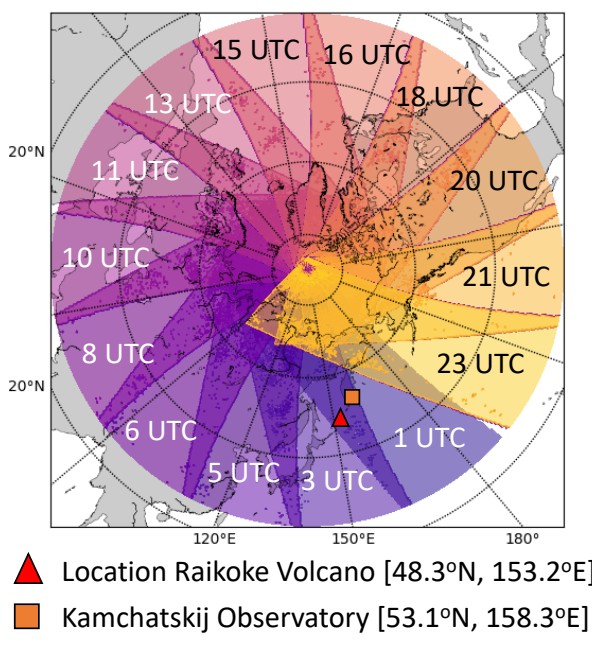

**Figure 1.** Example of daily TROPOMI overpasses (north of $25°$N, with a swath width of $2600$ km). Time indicates the approximate central time for each overpass (each track takes approximately 1.5-2 hours). Note the overlap of the swaths, resulting in a higher temporal resolution near the pole. Also shown are the location of the Raikoke volcano (triangle) and the radiosonde location at the Kamchatskij Airport (square).

The International Airways Volcanic Watch (IAVW) is responsible for the dissemination of information on the occurrence of volcanic eruptions and associated volcanic ash clouds (ICAO, 2019a) through nine Volcanic Ash Advisory Centres (VAACs). During a volcanic eruption the responsible VAAC disseminates relevant information to the aviation sector regarding the geographic location of volcanic ash present in the atmosphere. Currently, the VAACs are only required to provide forecasts of volcanic ash dispersion and therefore less development has been made on the forecasting of volcanic gas clouds. There is,

however, increasing consensus among the scientific community that monitoring and simulating $SO_2$ clouds could be of interest to stakeholders, as volcanic $SO_2$ can pose a public health hazard and potentially affect the aviation industry (e.g. increase of aircraft maintenance costs) (Witham et al., 2012; Schmidt et al., 2014; Carboni et al., 2016; Granieri et al., 2017; Grégoire et al., 2018). Volcanic $SO_2$ clouds are also frequently (but not always) co-located with ash clouds. Detecting ash clouds from satellites retrievals remains a challenging task and in some circumstances $SO_2$ clouds may act as proxies for ash clouds (e.g.,

Carn et al., 2009; Thomas and Prata, 2011; Sears et al., 2013; Kristiansen et al., 2015). As a result, the latest roadmap published by the IAVW (ICAO, 2019b) includes $SO_2$ forecasts as a core item to be implemented in the future.

    The main tools used by VAACs to provide accurate forecasts of volcanic cloud characteristics are atmospheric dispersion models (ADMs), which are numerical models that simulate how air parcels disperse within the atmosphere. ADMs are used for

a large variety of advection-related research, including dust transport, nuclear accidents, forest fires, air pollution, plant diseases and volcanic clouds (Carpenter et al., 2010; Webster et al., 2011; Katata et al., 2014; Schmidt et al., 2014, 2015; Ashfold et al., 2017; Meyer et al., 2017; Osborne et al., 2019). Because of the ash-focused task of VAACs, there has been a strong research focus on improving the simulation of volcanic ash in ADMs and measuring volcanic ash using in-situ and satellite measurement techniques (e.g., Witham et al., 2007; Corradini et al., 2011; Prata and Prata, 2012; Mulena et al., 2016; Harvey et al., 2018; Webster et al., 2020). The skill of ADMs in simulating the evolution of $SO_2$ clouds has also been investigated (e.g., Eckhardt et al., 2008; Heard et al., 2012; Boichu et al., 2013; Schmidt et al., 2015), but to a much lesser extent as this has generally been the realm of global climate models interested in the climatic impacts of the periodic stratospheric injections from volcanoes (e.g., Haywood et al., 2010; Solomon et al., 2011; Schmidt et al., 2018).

Observations are vital in determining the skill of the dispersion models. While in situ observations are available for several well-studied volcanoes (e.g., Pfeffer et al., 2018; Sahyoun et al., 2019; Whitty et al., 2020), they are only available for a limited number of locations. In recent decades, many high-resolution remote-sensing measurements have become available, providing a great data source on activity at even the most remote volcanoes across the globe. A large number of satellites now measure atmospheric $SO_2$, with each newly launched instrument having an increased accuracy (see for example fig. 2 in Theys et al. (2019)). The TROPOspheric Monitoring Instrument (TROPOMI) has been operational since the end of 2017 and retrieves atmospheric $SO_2$ total column densities at an unprecedented spatial resolution for UV measurements (up to $3.5 \, \mathrm{km} \times 5.5 \, \mathrm{km}$ at nadir) (Theys et al., 2017, 2019). TROPOMI therefore provides a useful source of information to evaluate model simulations of $SO_2$ clouds using ADMs (and climate models).

One major issue for the development of ADMs for volcanic clouds is the relatively small number of large-magnitude eruptions available since the start of the satellite era in 1979 in order to validate model output. While small-magnitude eruptions take place more frequently, large-magnitude eruptions that can emit large amounts of $SO_2$ into the stratosphere are much more sporadic (e.g., Pyle, 1995; Miles et al., 2004; Carn et al., 2016; Schmidt et al., 2018). The 2019 Raikoke eruption is the first eruption with $SO_2$ emissions in excess of $1 \, \mathrm{Tg}$ of $SO_2$ that has been observed by the TROPOMI instrument. Due to the amount of $SO_2$ emitted into the stratosphere, it provides an ideal test case to validate the skill of the Numerical Atmosphere-dispersion Modelling Environment (NAME) (Jones et al., 2007), which is the dispersion model used by the London VAAC (Witham et al., 2020). In this paper we will focus on the evolution of the volcanic $SO_2$ cloud during the first 3 weeks after the 2019 Raikoke eruption and compare the output from NAME with the TROPOMI and the Infrared Atmospheric Sounding Interferometer (IASI) satellite $SO_2$ products. In the accompanying Part 2 paper (Osborne et al., 2021), a detailed assessment of the sulfate aerosol together with volcanic ash from this eruption are discussed as well as the effects from biomass burning aerosols that were emitted into the stratosphere from an unusually strong pyrocumulus event in continental north America.

The manuscript is structured as follows: after discussing the TROPOMI and the IASI satellite $SO_2$ products in sections 2.1 and 2.2, we briefly introduce all the relevant aspects of the NAME dispersion model in section 2.3, including the eruption source parameters. Using the introduced input parameters for our 25-day long NAME simulations, we obtain a good qualitative comparison of the simulated $SO_2$ cloud with the TROPOMI satellite $SO_2$ products during the first 3 weeks after the eruption, which we present in section 3.1. A more detailed analysis of the model skill is presented in sections 3.2 and 3.3, where we show

that the Fractional Skill Score (FSS) and the Structure, Amplitude and Location (SAL) metric (both metrics are introduced in section 2.4) are powerful tools for assessing the skill of the model simulations in comparison to satellite measurements. The NAME simulation skill in terms of the NH-mean $SO_2$ mass burden throughout the first 25 days after the Raikoke eruption are presented in section 3.4, showing a large dependence of the mass burden evolution on the vertical emission profile. We finish with a discussion of our work (section 4) and present the main conclusions in section 5.

## 2 Data and Methods

### 2.1 TROPOspheric Monitoring Instrument

TROPOMI is part of the ESA's S5P satellite launched on 13 October 2017 (Veefkind et al., 2012) and is a polar-orbiting, sun-synchronous, hyperspectral spectrometer that measures Earth reflected radiances in the ultraviolet (UV), visible, near-infrared and shortwave infrared parts of the spectrum. Atmospheric $SO_2$ vertical column density (VCD, expressed in Dobson units with $1\,DU = 2.69 \times 10^{16}\,\mathrm{molecules\,cm^{-2}}$) is retrieved by applying Differential Optical Absorption Spectroscopy (DOAS) (Platt and Stutz, 2008) to the measured ultraviolet spectra in three wavelength ranges (312 nm - 326 nm, 325 nm - 335 nm, and 360 nm - 390 nm). For a more detailed description of the $SO_2$ retrieval, we refer the reader to Theys et al. (2017) and Theys et al. (2018).

In our study we use the TROPOMI satellite retrievals across the NH (north of $25^{\circ}$ N) that cover the Raikoke $SO_2$ cloud between 22 June and 15 July 2019. Figure 1 shows an example of the TROPOMI daily overpasses over the NH. Compared to its predecessors OMI and SCIAMACHY, TROPOMI has a higher horizontal pixel resolution (up to 3.5 km $\times$ 5.5 km), allowing for a more detailed characterisation of the small-scale features in volcanic $SO_2$ clouds (Theys et al., 2019). The retrieved TROPOMI $SO_2$ VCD product is calculated by accounting for a large number of parameters, such as meteorological cloud fraction, surface albedo and the vertical distribution of absorbing trace gases (e.g. ozone) (Theys et al., 2017, 2018). As a result, the $SO_2$ retrievals from TROPOMI are sensitive to many assumptions which can lead to uncertainties of up to $\pm50\%$. For $SO_2$ in the stratosphere, the sum of the various uncertainties can be approximated to be around $\pm30$ % of the retrieved $SO_2$ VCDs. For a detailed discussion of the retrieval uncertainties see Theys et al. (2017).

One of the largest uncertainties of the TROPOMI $SO_2$ VCD product is the height of the $SO_2$ cloud. However, the sensitivity of the measurement with height is well characterised. To account for this sensitivity, the TROPOMI $SO_2$ VCD level 2 products are publicly-available from the ESA website (https://s5phub.copernicus.eu, last access: 22 March 2021) for three different scenarios where the $SO_2$ layer is assumed to be at either 1 km, 7 km or 15 km above sea level (asl). The TROPOMI VCD data presented in this study are for an $SO_2$ layer at 15 km asl, as this is the height nearest to the estimated weight-averaged emission height for the Raikoke eruption (see e.g. figure 2).

In order to compare the available satellite retrievals with any atmospheric dispersion model output, one needs to apply the column averaging kernel (AK) operators to the model data, thereby matching the model-simulated $SO_2$ VCDs to the TROPOMI products. The pre-calculated AKs for the 15 km scenario (Theys et al., 2018) have been applied to the NAME model output.

We have repeated the analysis using the AKs and TROPOMI VCDs assuming the $SO_2$ layer at 7 km asl, which affects the absolute $SO_2$ VCDs (not shown), but not our interpretation of the results or our overall conclusions.

To obtain a daily $SO_2$ mass estimate from the TROPOMI measurements, we grid the satellite data and combine all the overpasses during a 24-hour period starting at 12 UTC of any given day. In the case of multiple overpasses over a single location, we average the $SO_2$ cloud at these grid locations to avoid double counting. For the mass estimate from TROPOMI we have used a detection threshold of 0.3 DU (Theys et al., 2020). The resulting $SO_2$ VCD (in DU) is then converted into a mass (Tg) by using the area of each individual grid point and the molar mass of $SO_2$. Due to the high spatial resolution of the TROPOMI retrieval (on average nine TROPOMI pixels per output grid cell for the resolution used in the dispersion model), we have down-scaled the final TROPOMI retrievals to the output grid resolution of the NAME dispersion model by averaging the $SO_2$ VCDs of all pixels within each NAME grid cell ($0.2°$latitude $\times$ $0.4°$longitude). Unless otherwise specified, we refer to the sulfur dioxide mass burden as the total $SO_2$ mass within the NH, north of $25°$ N.

During the initial stage of the eruption, it is likely that TROPOMI underestimates the $SO_2$ VCDs due to the presence of volcanic ash (e.g., Yang et al., 2010). To understand if ash interference is likely to have affected our $SO_2$ estimates, we also retrieve the absorbing aerosol index (AAI) from the TROPOMI instrument (Zweers, 2016). Although the TROPOMI AAI product should be used with care due to its sensitivity to e.g. cloud height and optical thickness (see e.g., de Graaf et al., 2016; Kooreman et al., 2020), high index values ($> 1$) indicate the presence of aerosol plumes from dust outbreaks, volcanic ash, and biomass burning. During the first 48 hours after the eruption we found high peak AAI values within the volcanic cloud ($> 18.9$ on the 22 June and $> 3.5$ on the 23 June), indicating that volcanic ash had an impact on the $SO_2$ retrieval during this period.

## 2.2 The Infrared Atmospheric Sounding Interferometer

The second satellite $SO_2$ dataset used in our analysis is acquired using IASI onboard the Metop-A and Metop-B satellites. These satellites operate in tandem on a polar orbit with a field-of-view (FOV) consisting of four circular footprints of 12 km diameter (at nadir) inside a square of 50 km $\times$ 50 km and provide a global coverage twice a day. For our analysis we use the $SO_2$ plume height estimates based on the IASI data, which is produced by applying the retrieval algorithm presented in Carboni et al. (2012) and Carboni et al. (2016). The IASI instrument also retrieves the $SO_2$ VCDs within the volcanic plume, but uses a different set of assumptions in the retrieval algorithm compared to TROPOMI (e.g. IASI retrieves the plume height which effects the $SO_2$ VCD. The retrieved IASI plume height can be different from the plume heights assumed in the TROPOMI product and therefore the $SO_2$ VCDs from the two methods are not equivalent). To compare $SO_2$ VCDs from NAME to the IASI data, one would therefore also need to apply a different scaling (i.e. AK). As the TROPOMI and IASI retrieval assumptions and limitations are satellite-specific (for example, TROPOMI might detect $SO_2$ closer to the surface than IASI due to the presence of water vapour), a comparison between the two $SO_2$ VCD products is not straightforward and is not attempted here. While it would be an interesting exercise to apply our analysis also to the IASI data, we focus on the comparison of NAME with the TROPOMI $SO_2$ estimates and therefore no further analysis is done for the IASI $SO_2$ VCD retrievals.

## 2.3 Numerical Atmosphere-dispersion Modelling Environment (NAME)

The Numerical Atmospheric-dispersion Modelling Environment (NAME) is a Lagrangian model developed by the Met Office (Jones et al., 2007) and is the operational dispersion model used by the London VAAC to forecast the dispersion of volcanic clouds within European Airspace (e.g. the Icelandic eruptions of Eyjafjallajökull in 2010 and Holuhraun in 2014-2015). For our work we use NAME version 8.1. The model can trace both ash particles and gases through the atmosphere and includes chemistry parameterisations that allow the conversion of $SO_2$ into sulfate aerosols ($SO_4$) within the volcanic cloud (see section

2.3.3). There is no radiative nor chemical interaction between the ash particles and the sulfur species in NAME; the ash particles and sulfate aerosols are thus considered to be externally mixed. In this section we focus on the dispersion of $SO_2$ and highlight the important aspects of NAME for this part of the research. More details on the modelling of ash particles within the model are discussed in the accompanying Part 2 paper (Osborne et al., 2021).

### 2.3.1 Simulating volcanic cloud dispersion using NAME

Simulating the dispersion of a volcanic cloud with NAME relies on the tracing of air parcels through the atmosphere, each containing an ash, $SO_2$ and/or $SO_4$ mass. These air parcels are released from the 'source' location (volcano), where the user has to define the eruption source parameters (see section 2.3.4). NAME is an offline model, therefore each parcel is advected by an externally obtained wind field (e.g. a high resolution Numerical Weather Prediction (NWP) model). In our simulations, we use the wind fields from the latest global analysis of the Met Office Unified Model (MetUM), which have a horizontal resolution

of around 10 km at mid-latitudes, 59 levels between the surface and 30 km asl (decreasing vertical resolution with altitude, with approximately 600 m resolution at tropopause height) and a 3 hourly temporal resolution. The path of each trajectory is calculated using the following equation:

$$\boldsymbol{x}(t + \Delta t) = \boldsymbol{x}(t) + [\boldsymbol{u}(\boldsymbol{x}(t)) + \boldsymbol{u}'(\boldsymbol{x}(t))]\Delta t, \tag{1}$$

where $\boldsymbol{x}(t)$ is the location of the parcel at time $t$, $\boldsymbol{x}(t+\Delta t)$ the new location of the parcel at time $t+\Delta t$, $\boldsymbol{u}(\boldsymbol{x}(t))$ the 3D-wind

vector at location $\boldsymbol{x}(t)$ and $\boldsymbol{u}'(\boldsymbol{x}(t))$ represents a stochastic perturbation to the parcels trajectory representing turbulence and unresolved sub-grid mesoscale wind variations in the dispersion model. In NAME, $\boldsymbol{u}'$ consists of two parts representing atmospheric turbulence ($\boldsymbol{u}'_{turb}$) and sub-grid mesoscale diffusion ($\boldsymbol{u}'_{meso}$). The turbulence part represents the stochastic motions from the air parcels due to small-scale perturbations. The mesoscale diffusion represents the horizontal mesoscale motions in the atmosphere that are not captured by the resolution of the used NWP model. Each NWP model has a limited spatial

and temporal resolution and as a result, part of the mesoscale features (e.g. eddies) are not captured by the NWP wind field provided. Both the turbulence and mesoscale diffusion within the free atmosphere (excluding the Planetary Boundary Layer, which has a more detailed scheme (Webster et al., 2018)) are calculated using:

$$\boldsymbol{u}'_{turb} \cdot \Delta t = d\sqrt{2\,\boldsymbol{K_{turb}} \cdot \Delta t}, \tag{2}$$

$$\boldsymbol{u}'_{meso} \cdot \Delta t = d\sqrt{2\,\boldsymbol{K_{meso}} \cdot \Delta t}, \tag{3}$$

$$\boldsymbol{K_x} = (\sigma_u^2 \tau_u, \sigma_v^2 \tau_v, \sigma_w^2 \tau_w), \tag{4}$$

**Table 1.** The values for the diffusion parameter $K$ used in NAME. Values are NWP dependent and are given here for the Met Office Unified Model Global analysis (10 km horizontal resolution, 59 levels) with a 3 hourly temporal resolution. Values for $\sigma^2$ are given instead of $\sigma$ to be consistent with the values presented by Webster et al. (2018). Values are given for both the turbulence $K_{turb}$ and the mesoscale diffusion $K_{meso}$ that are used in NAME for the free atmosphere (i.e. excluding the boundary layer).

| Global MetUM analysis ($0.140625° \times 0.09375°$, 59 levels, 3 hourly resolution) | | | | | |
|---|---|---|---|---|---|
| | $K$ (m$^2$s$^{-1}$) | $\sigma_{u,v}^2$ (m$^2$s$^{-2}$) | $\tau_{u,v}$ (s) | $\sigma_w^2$ (m$^2$s$^{-2}$) | $\tau_w$ (s) |
| $K_{turb}$ | (18.75,18.75,1) | 0.0625 | 300 | 0.01 | 100 |
| $K_{meso}$ | (6400,6400,0) | 0.64 | 10000 | 0 | 0 |

where $K_x$ is a 3D-diffusion vector defined separately for both components, using typical values for the standard deviation of the velocity ($\sigma$) and typical time length-scales ($\tau$). $d$ represents a random number from a top-hat distribution within the range [-1,1]. The values for $\sigma$ and $\tau$ are dependent on the NWP model used, as they are impacted by the resolution of the model (Harvey et al., 2018; Webster et al., 2018). The values for $\sigma$ and $\tau$ used in this study are obtained from the analysis done by Webster et al. (2018) and are shown in table 1. Note that for $K_{meso}$ the vertical component ($\sigma_w^2 \tau_w$) is zero.

Accurately describing atmospheric dispersion due to mixing is a complex three dimensional problem (e.g., Waugh et al., 1997; Haynes and Anglade, 1997; Haynes and Shuckburgh, 2000; Hegglin et al., 2005; Wang et al., 2020). TROPOMI satellite retrievals provide $SO_2$ VCDs and therefore our information on mixing effects of the $SO_2$ cloud is limited to their horizontal impact. Studies by Balluch and Haynes (1997) and Haynes and Anglade (1997) have shown that the vertical and horizontal components of stratospheric mixing are related, which allowed them to derive an 'effective horizontal' diffusion from observations. Values reported in the literature for horizontal diffusion coefficients in the lower stratosphere vary over an order of magnitude ($10^3$ m$^2$s$^{-1}$ - $10^5$ m$^2$s$^{-1}$) and also depend on the resolution of the NWP data used (e.g., Balluch and Haynes, 1997; Waugh et al., 1997; Harvey et al., 2018; Wang et al., 2020). To investigate the importance of the horizontal mesoscale diffusion parameter, we present two sensitivity simulations with two different $SO_2$ emissions profiles (see section 2.3.4 for discussion of these profiles) and a reduced value for $K_{meso}$ (see table 2). We decided to only change the $K_{meso}$ parameter, as the horizontal $K_{turb}$ components (see table 1) are at least an order of magnitude smaller than the $K_{meso}$ components and thus changing them would not show any significant impact on our initial results that is not captured by changing $K_{meso}$. The simulations with a reduced value for $K_{meso}$ are indicated by the subscript $_{rd}$ throughout this study. Due to the large range of potential realistic $K_{meso}$ values, we have stepwise reduced the parameter and found the best results for a 75% reduction, which is the value presented in this paper and is similar to the values reported by Balluch and Haynes (1997) and Waugh et al. (1997).

### 2.3.2 Calculating $SO_2$ mass estimates from NAME

In our simulations, the $SO_2$ concentrations (kg m$^{-3}$) from all the individual air parcels in NAME are presented as hourly-means on a regular latitude-longitude grid by calculating the total mass of the $SO_2$ of all parcels in each grid box every hour. The NAME output is calculated using a grid size of $0.2°$ latitude and $0.4°$ longitude (approximately 20 km $\times$ 20 km at the

latitude of the Raikoke volcano). The vertical resolution of the output is 500 m up to 15 km asl and 1 km resolution up to 20 km asl, giving a total of 35 levels.

To compare the daily $SO_2$ mass estimates from NAME and TROPOMI, we select the hourly NAME output corresponding to each individual TROPOMI overpass time and select only the grid boxes in NAME that are in the domain scanned by TROPOMI during that overpass. To calculate the $SO_2$ VCD, we apply the corresponding column AK operators obtained from TROPOMI (see section 2.1) to each grid cell of the NAME output. Then for each column on the NAME output grid, the $SO_2$ concentrations ($kg\,m^{-3}$) in each grid cell are vertically integrated to obtain the $SO_2$ VCD estimate from NAME.

In all our NAME simulations we found that the $SO_2$ cloud is more diffuse than observed by TROPOMI. Therefore, removing all $SO_2$ VCDs <0.3 DU in NAME (which is the detection threshold we have used for TROPOMI, see section 2.1) from the simulations would result in a negative bias within the NAME simulation mass estimates that are not related to the evolution of the cloud, but due to the stronger diffusion within the model. Therefore we have not included a detection threshold when determining the $SO_2$ mass estimates for the NAME simulations. Similarly to the TROPOMI estimate, the daily mass estimates from NAME are calculated during a 24-hour period starting at 12 UTC on any given day. The $SO_2$ mass burden is defined as the total $SO_2$ mass (Tg) within the NH, north of $25^\circ$ N.

### 2.3.3 Chemistry within NAME

NAME contains an atmospheric chemistry scheme (Redington et al., 2009). The relevant chemistry for volcanic clouds is related to the conversion of $SO_2$ into sulfate ($SO_4^{2-}$). NAME accounts for the oxidation of $SO_2$ in the gas phase using the following reaction

$$OH + SO_2 + M \rightarrow HSO_3 + M \tag{R1}$$

where $HSO_3$ is then rapidly oxidised to $H_2SO_4$ on formation. When water is present in the atmosphere, the oxidation can happen in the aqueous phase by both $H_2O_2$ and $O_3$ through the following reaction:

$$SO_2 + H_2O \rightleftharpoons H^+ + HSO_3^- \tag{R2}$$
$$HSO_3^- \rightleftharpoons H^+ + SO_3^{2-} \tag{R3}$$

which is followed by

$$HSO_3^- + H_2O_2 \rightarrow SO_4^{2-} + H^+ + H_2O \tag{R4}$$
$$HSO_3^- + O_3 \rightarrow SO_4^{2-} + H^+ + O_2 \tag{R5}$$
$$SO_3^{2-} + O_3 \rightarrow SO_4^{2-} + O_2 \tag{R6}$$

Reactions R2-R6 dominate in cloudy conditions and only occur in model grid boxes when both the meteorological cloud fraction and liquid water content are non-zero. The concentrations of $H_2O_2$ and $O_3$ in the atmosphere are pre-defined in the NAME model by using monthly-mean background fields obtained from a historical 'Unified Model coupled to the United

**Table 2.** Overview of the NAME simulations performed using different emission profiles and a reduced mesoscale diffusion (values for $K_{meso}$ can be found in table 1). Also shown is the estimated mass emitted into the stratosphere. For the actual vertical emission profiles, please see fig. 2. All the simulations use the same NWP data input (Global MetUM), the same emission location (48.3° N, 153.2° E), emission duration (21 June 1800 UTC - 22 June 0300 UTC 2019), simulation domain (NH between 25-90°N) and simulation length (25 days).

| Simulation name | Mass emitted | Profile used | Mass Stratosphere | Mesoscale diffusion |
|---|---|---|---|---|
| VolRes1.5 | 1.5 Tg | VolRes1.5 | 0.64 Tg | $K_{meso}$ |
| VolRes2.0 | 2.0 Tg | VolRes2.0 | 0.85 Tg | $K_{meso}$ |
| StratProfile | 1.57 Tg | StratProfile | 1.09 Tg | $K_{meso}$ |
| VolRes1.5$_{rd}$ | 1.5 Tg | VolRes1.5 | 0.64 Tg | $0.25\,K_{meso}$ |
| StratProfile$_{rd}$ | 1.57 Tg | StratProfile | 1.09 Tg | $0.25\,K_{meso}$ |

Kingdom Chemistry and Aerosol model' (UM-UKCA model) simulation that have been smoothed between months using interpolation.

In NAME the $SO_2$ and sulfate aerosol particles can be removed through dry and wet deposition. For our simulations we found that the dry deposition had limited importance for the 2019 Raikoke eruption as most of the volcanic clouds are at high altitudes. Wet deposition in NAME is calculated using a standard depletion equation:

$$\frac{dC}{dt} = \Lambda C \tag{5}$$

$$\Lambda = Ar^B \tag{6}$$

with C representing the $SO_2$ concentration $(\mathrm{kg\,m^{-3}})$ and $\Lambda$ the scavenging coefficient, which is calculated based on the rainfall rate $r$ (in mm/hr) and two scavenging parameters A and B. The parameters A and B vary for different types of precipitation (i.e., large-scale/convective and rain/snow) and for different wet deposition processes (i.e., rainout, washout and the seeder-feeder process). For more detailed information, including the specific values for A and B for $SO_2$ and sulfate aerosols, we refer to Webster and Thomson (2014), Leadbetter et al. (2015) and references therein.

### 2.3.4 Eruption Source Parameters

When simulating a volcanic eruption, the NAME dispersion model needs eruption source parameters (ESP) consisting of 1) location, 2) timing, 3) mass eruption rate $(\mathrm{kg\,s^{-1}})$ and 4) vertical emission profile, and for simulating volcanic ash also 5) particle density, shape and particle size distribution. Here we will discuss ESP 1-4. For information about the setup of the simulations including ash, we refer to the Part 2 paper (Osborne et al., 2021). For all simulations described in this manuscript, we released a total of 10 million air parcels in NAME within a column above the volcano, each parcel representing an equal amount of $SO_2$ mass. All simulations are run for 25 days until 15 July 2019 and the simulation domain is the NH, north of 25° N).

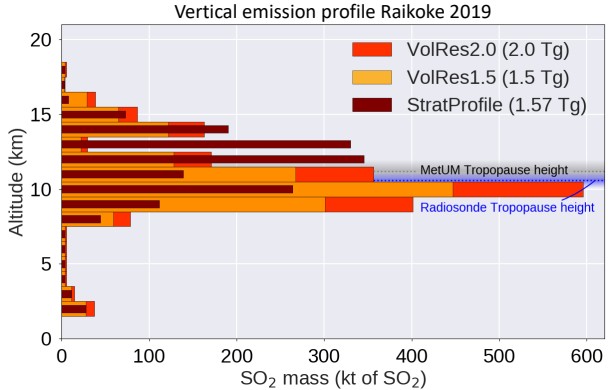

**Figure 2.** Estimated total emitted $SO_2$ mass for the Raikoke eruption between 21 June 1800 UTC and 22 June 0300 UTC 2019. The initial emission profile was provided by the VolRes (Volcano Response) team, which is implemented in NAME for the 1.5 Tg $SO_2$ simulation (VolRes1.5, orange). We also simulated the same profile for a 2.0 Tg $SO_2$ emission (VolRes2.0, red). Also included is a new emission profile estimate (StratProfile, brown) based on the TROPOMI $SO_2$ VCD cloud on 23 June (see section 2.3.5). This profile has a similar total mass emitted to VolRes1.5, but a larger fraction (69% instead of 43%) of its mass is emitted in the stratosphere (see table 2). The grey line represents the average tropopause height in the MetUM during the first 36 hours after the eruption at the location of Raikoke volcano. The blue line and shading represents the average and the range of measured tropopause heights by the radiosondes released from the Kamchatskij Observatory (square in fig. 1) during the first 36 hours after the eruption.

In all our simulations (see table 2 for overview), we release the $SO_2$ from the location of the volcano (48.3° N, 153.2° E) between 21 June 18 UTC and 22 June 03 UTC. The timing of the $SO_2$ release is in line with the source term provided by the Volcano Response (VolRes) team (https://wiki.earthdata.nasa.gov/display/volres). The VolRes team is an international research collaboration to coordinate a response plan after large volcanic eruptions using observational and modelling tools. No information on the temporal variation in the mass eruption rate was provided by VolRes, thus we assume a constant mass eruption rate throughout the entire eruption period.

The Raikoke eruption injected most $SO_2$ mass near the tropopause height (see figure 2), but the precise emission profile is uncertain (e.g. Hedelt et al., 2019; Kloss et al., 2021). Small changes in the $SO_2$ emission profile could lead to a large change in the amount of mass emitted into the stratosphere, which will strongly influence the evolution of the $SO_2$ cloud. Therefore, in our study we use three different $SO_2$ emission profiles that vary in terms of the $SO_2$ mass that is emitted into the stratosphere as shown in fig. 2. The VolRes1.5 $SO_2$ emission profile is based on the vertical mass distribution obtained from the VolRes team using IASI retrievals on 22 June, as shown by the orange bars in fig. 2. The total $SO_2$ mass emitted based on the VolRes estimate, which is also the mass emission used in the VolRes1.5 profile, was approximately $1.5 \pm 0.2$ Tg of $SO_2$.

To determine what fraction of the total $SO_2$ mass was emitted into the stratosphere, we calculated the tropopause height in the MetUM global analysis using the World Meteorological Organization (WMO) temperature lapse rate definition. Using the spread in the 150 nearest grid points to the volcano location in the model, we get an average tropopause height of $11.2 \pm 0.7$ km during the first 36 hours after the eruption. To verify this tropopause height, we used radiosonde data from the Kam-

chatskij Observatory (see fig. 1), which is the nearest radiosonde location to the Raikoke volcano (data can be retrieved from http://weather.uwyo.edu/upperair/sounding.html). Using the same tropopause height criteria for the radiosondes released from this location, we estimate an average tropopause altitude of $10.5 \pm 0.7$ km during the first 36 hours, showing that the MetUM simulated the tropopause height within the expected range. Using the MetUM tropopause height estimate, the VolRes1.5 profile emits 0.86 Tg into the upper troposphere (UT) with a peak at 10 km altitude and emits 0.64 Tg into the lower stratosphere (LS, defined as the layer between the tropopause and 18 km asl) with a secondary peak at 14 km asl.

In the case of a multi-phase plume like Raikoke (multi-phase here refers to the mixture of ash, sulfate aerosols and gas present in the cloud, not the number of eruption phases), high ash concentrations within the volcanic cloud can interfere with satellite $SO_2$ retrievals (Yang et al., 2010; Carboni et al., 2016; Theys et al., 2017) leading to an underestimation of the $SO_2$ VCDs. Furthermore it is known that, in the stratosphere, ash and sulfate aerosols can have a local heating effect due to their interactions with radiation, resulting in lofting of the $SO_2$, sulfate aerosols and ash (e.g., Niemeier et al., 2009; Jones et al., 2016; Muser et al., 2020). NAME does not account for radiative lofting of volcanic species due to changes in heating rates as it is an offline model driven by NWP wind fields that are not affected by any volcanic ash or aerosols radiative effects.

The fact that Raikoke was an eruption that produced a multi-phase plume that emitted 1.5 Tg of $SO_2$ and 15 Tg of ash (Osborne et al., 2021) near the tropopause (see figure 2), justifies simulations using different initial $SO_2$ emission profiles and warrants a closer investigation of the emission profile provided by the VolRes team. To understand potential uncertainties on the VolRes emission profile, we have run an initial 36-hour NAME simulation with the VolRes1.5 vertical emission profile input and compared the $SO_2$ VCD estimates (fig. 3a) with the TROPOMI retrieval on 23 June (fig. 3b). The comparison reveals that the VolRes1.5 simulation has a different longitudinal distribution compared to the TROPOMI satellite retrievals. Figure 3d shows the averaged $SO_2$ VCDs between 48-52°N along section I-II (black box in fig. 3a) for the clouds shown in panels a-c. This shows that along the northern part of the cloud between 170-175°E, the VolRes1.5 simulation underestimates the $SO_2$ VCDs from TROPOMI by up to a factor of 8.

Figure 4 shows the vertical cross section from the VolRes1.5 simulation through the $SO_2$ cloud along section I-II in fig. 3a, together with the available estimated cloud heights from IASI for all pixels between 49-50°N. The cloud height from the IASI retrieval is estimated using the method described in Carboni et al. (2016). Figure 4 shows that the $SO_2$ cloud between 170-175°E is simulated in NAME between 11-14 km, which coincides with the UT/LS in the MetUM Global model (see fig. 2). The altitude of the $SO_2$ cloud for the NAME VolRes1.5 simulation along the cross section shown in fig. 4 is within the uncertainty range of the IASI height estimates and gives us confidence that the NAME simulated cloud height range is realistic. However, the underestimated $SO_2$ VCDs in the latitude range 170-175°E for the VolRes1.5 simulation compared to TROPOMI (fig. 3) could indicate an underestimation of the mass fraction of $SO_2$ present in the stratosphere, which could be due to the lack of radiative lofting of the $SO_2$ during the first 36 hours after the eruption. The study by Muser et al. (2020) shows that after the Raikoke eruption, most of the radiative lofting of the ash layers occurred over these timescales and we assume similar timescales to be applicable for the lofting of the $SO_2$ clouds. Reducing the horizontal diffusion $\boldsymbol{K_{meso}}$ in the VolRes1.5$_{rd}$ simulation (not shown) resulted in a simlar underestimate as shown in figure 3d for VolRes1.5, excluding that over dispersion in the stratosphere is the main source for the underestimate.

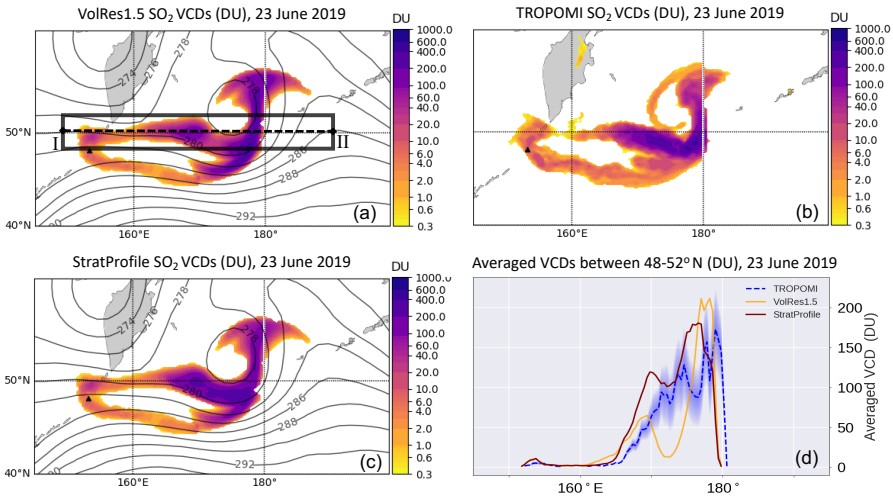

**Figure 3.** The $SO_2$ VCD estimates for 23 June 2019 for a) the VolRes1.5 simulation, b) the TROPOMI retrievals and c) the StratProfile simulation. The TROPOMI retrievals are downscaled to the NAME simulation resolution, (i.e. averaged per grid box, see sections 2.1). The black contours show the pressure at the 10 km asl in the MetUM analysis used for both NAME simulations. Panel d shows the latitudinal $SO_2$ VCDs from panels a-c along section I-II in fig. 3a, averaged over the black box between $48°N$ and $52°N$. Shading represents the standard error estimate for the TROPOMI estimate.

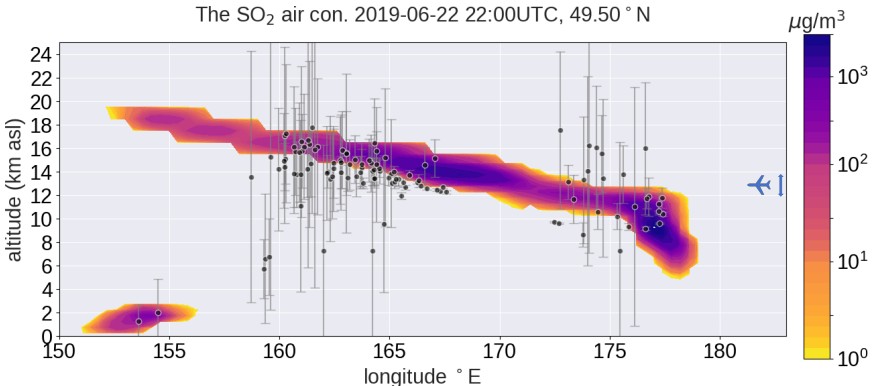

**Figure 4.** Vertical cross section of $SO_2$ mass concentrations ($\mu g/m^3$) in the VolRes1.5 simulation and the IASI height estimate along the line I-II in fig. 3a on 22 June 2019 at 22 UTC. This time corresponds with the IASI overpass over the volcanic cloud, thereby minimising displacement errors due to timing. The black dots represent the available height estimates including error bars from the IASI retrieval of the $SO_2$ cloud for all pixels between $49-50°N$ and is estimated following the method described in Carboni et al. (2016). The blue arrow on the right of the figure indicates the range of cruise altitudes for long-haul aircraft (11.9-13.7 km).

Based on the initial findings from the VolRes1.5 simulation, we also conduct a simulation in which we released a total of 2 Tg of $SO_2$, using the same relative mass distribution in the vertical, termed VolRes2.0. The experiment emits a larger amount

of $SO_2$ into the LS (1.15 Tg in the UT and 0.85 Tg in the LS). In chemistry transport models (including NAME), the chemical conversion and the rate of wet and dry deposition of $SO_2$ depends on the $SO_2$ concentration. Therefore this simulation is not a simple scaling of the VolRes1.5 results.

In addition, we derive a different vertical profile based on the TROPOMI $SO_2$ VCD retrievals (StratProfile, for derivation see Appendix A) in which we use a different relative mass distribution in the vertical. In contrast to the VolRes1.5 profile (which

is based on the IASI satellite overpasses on 22 June), our StratProfile emission profile is based on the 23 June overpasses of TROPOMI. These overpasses are approximately 30 hours after the onset of the eruption and show a reduced ash interference (as seen by the strongly reduced AAI values, see section 2.1) thus we expect the $SO_2$ retrievals to be more accurate. Furthermore, this 'effective' emission profile (Rix et al., 2009; Klüser et al., 2013) will take into account any lofting of the $SO_2$ clouds resulting from the radiation interactions which may be enhanced by the presence of ash during the first 30 hours after the

eruption. The derived StratProfile emission profile releases similar amounts of $SO_2$ into the atmosphere as the VolRes1.5 run (1.57 Tg), but has the main peak in $SO_2$ mass at 12-13 km altitude. As a result, the StratProfile emits a much higher fraction (69% or 1.09 Tg) of the mass into the LS (VolRes emission profile emits 43% or 0.64 Tg into the LS).

## 2.4 Metrics to determine the skill of the NAME simulations

Assessing the model's skill in representing satellite measurements of $SO_2$ requires appropriate metrics. Similar comparisons

should be possible in almost near-real time for VAACs when investigating future eruptions. Therefore, apart from being able to show the details of the model-satellite comparison, it is also important that the metric is easily interpretable by end-users. In the following sub-sections we introduce two metrics for identifying the skill of the simulations: (1) the FSS and (2) the SAL-score.

### 2.4.1 Fractional Skill Score (FSS)

The FSS was originally developed to determine the skill of weather forecast models to represent radar rainfall observations

(Roberts, 2008; Roberts and Lean, 2008; Mittermaier et al., 2013), but has been since used to also describe the skill of dispersion models in representing volcanic clouds (Dacre et al., 2016; Harvey and Dacre, 2016). For volcanic $SO_2$ clouds, the FSS is calculated using the ratio between the model-simulated ($M_k$) and observed ($O_k$) fractional coverage of the $SO_2$ cloud at each location (neighbourhood) in the domain investigated. When considering a neighbourhood of $N$ grid points (or pixels), the FSS is calculated using:

$$FSS = 1 - \frac{FBS}{FBS_{ref}} \tag{7}$$

$$FBS = \frac{1}{N} \sum_{k=1}^{N} (O_k - M_k)^2 \tag{8}$$

$$FBS_{ref} = \frac{1}{N} \left[ \sum_{k=1}^{N} O_k^2 - \sum_{k=1}^{N} M_k^2 \right] \tag{9}$$

The FSS is calculated from the Fractions Brier Score (FBS), which is a variation on the Brier Score (Brier, 1950), and FBS$_{ref}$ is the largest FBS score one can obtain from multiple non-zero fractions within the domain when there is no overlap

between the two fields. In the case that observations and simulation are perfectly aligned, FSS is equal to one. In the case of a total mismatch FSS equals to zero. In general for the FSS, a model-simulation is considered to have skill when FSS > 0.5 (see e.g., Harvey and Dacre, 2016).

The FSS metric is very suitable for studying the skill of a model in capturing the volcanic cloud's spatial extent. One advantage of using the FSS metric is that it relaxes the requirement for exact matching of the spatial features in the simulations

with the observations. Instead when the fractional coverage of the SO$_2$ cloud within a studied region (i.e. a neighbourhood of size N) is the same for the observations and the simulation, this metric counts it as a correct forecast. By using different sizes of neighbourhoods, one can also determine at which spatial resolution the simulation is skilful (i.e. for which N is FSS > 0.5) at any given time, which helps to determine at which spatial scale features of the SO$_2$ cloud can be considered realistic. However the method does not consider differences in SO$_2$ VCDs values – it only considers a 'hit' or 'miss' for each location.

By applying the same FSS metric to increasing SO$_2$ VCD thresholds (i.e. sub-regions of the cloud), one can obtain information about model skill at simulating volcanic SO$_2$ cloud structures with varying vertical column densities.

### 2.4.2 Structure, Amplitude and Location Score (SAL-score)

The SAL-score is a metric that is composed of three components, which describe the Structure (S), Amplitude (A) and Location (L) of an investigated feature within a specified domain. The metric was originally developed to compare the structure of model-

simulated precipitation fields with observations (Wernli et al., 2008), but has since been adapted to also describe other fields, including volcanic clouds (e.g., Dacre, 2011; Wilkins et al., 2016; Radanovics et al., 2018). Here we will adopt this metric to describe the evolution of the SO$_2$ cloud. For a detailed description of the equations used to calculate each individual component we refer the reader to Wilkins et al. (2016).

Briefly, to calculate the S and L scores (not needed for the A score), we have to identify all the individual simulated and

observed SO$_2$ clouds. In our analysis each cloud is identified as a group of adjacent grid cells which have a SO$_2$ VCD value above a certain threshold. From Theys et al. (2019) we deduce that the detection limit of the satellite measurements for individual pixels is approximately 1 DU. All of the analysis in our study is done at the highest resolution that is available for all fields, which is the NAME model output (0.2°latitude and 0.4°longitude). Due to the higher spatial resolution of the satellite product, we have to average the TROPOMI output of multiple pixels within each NAME grid box (on average 9 TROPOMI

pixels per NAME output grid box at each given time step) to get both datasets on the same output grid. As a result, we have used a lower detection threshold of 0.3 DU when identifying all grid points that are part of a SO$_2$ cloud for the (re-gridded) TROPOMI retrievals and the NAME simulations. To remove additional spurious data from the TROPOMI satellite product, we also include a minimum size of each identified SO$_2$ cloud to be 100 km$^2$ (approximately the NAME grid box size at 50°N) before considering in our analysis. Simulated and observed SO$_2$ VCD values below either of these thresholds are excluded

from all S, A, and L calculations. SAL scores have been calculated by comparing the SO$_2$ VCD estimates from each NAME

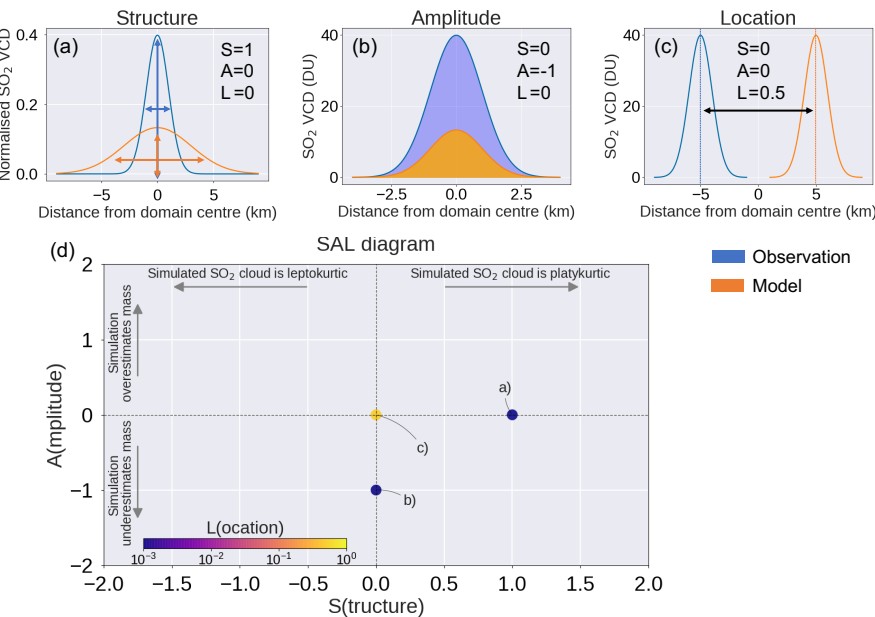

**Figure 5.** Schematic overview of the SAL score and its interpretation, using two cross-sections of idealised gaussian-shaped $SO_2$ clouds. Each panel shows the impact of an individual component of the SAL-score: a) Structure, b) Amplitude and c) Location (only the $L_1$ part). A negative S score indicates that the simulated $SO_2$ clouds are too narrow or have too high peak $SO_2$ VCD values when compared to the observed cloud (leptokurtic). When the simulated $SO_2$ clouds are too wide spread or have too low peak $SO_2$ VCD values, this is indicated by a positive S score (platykurtic). Panel d) shows an example of the SAL-score diagram with the scores of the three cases in a), b) and c) included. The horizontal axis represents the S score, the vertical axis the A score and the colour of each point represents the L score. When the simulation and observations compare perfectly, the score of each of the components is 0. The simulation and observations compare best when all the points are near the origin and have the dark purple colour.

simulation with the individual TROPOMI overpasses, as well as the daily averages. When calculating the individual overpass SAL-score values, we only included the NAME simulation data within the region covered by the TROPOMI overpass.

To interpret the SAL-score, we first assume a single idealised 2D-gaussian shaped cloud for both the simulated and observed $SO_2$ VCDs. Looking at the schematic cross-section presented in fig. 5, three characteristics are represented by the S, A and L 395 scores. The S score compares the shapes of each individual $SO_2$ cloud in terms of the horizontal extent (width) and maximum concentrations within the cloud, by comparing the normalised shape of the clouds (i.e. total mass of the simulated and the observed clouds are made equal, see fig. 5a). A negative S score indicates that the simulated $SO_2$ clouds are too narrow or have too high peak $SO_2$ VCD values when compared to the observed cloud (leptokurtic). When the simulated $SO_2$ clouds are too wide spread or have too low peak $SO_2$ VCD values, this is indicated by a positive S score (platykurtic).

The A score represents the comparison between the simulated and the observed total mass of $SO_2$ within the entire studied domain and is independent on the number of individual $SO_2$ clouds. Negative A scores represent an underestimate of the total

SO$_2$ mass in the simulation when compared to the observations (fig. 5b), while a positive value shows that the simulation is overestimating the total SO$_2$ mass in the domain.

Finally the L-score represents the distribution of the individual simulated and observed SO$_2$ clouds within the domain and consists of two parts L$_1$ and L$_2$ (Wernli et al., 2008). L$_1$ represents the normalised distance between the domain-averaged centre of mass of all the simulated and observed SO$_2$ clouds, where a higher positive value represents a larger distance between the simulated and observed domain-averaged centres of mass (see fig. 5c). In the case of multiple SO$_2$ clouds, L$_2$ represents the differences in the distribution of individual clouds around the domain-averaged total centre of mass. L$_2$ is calculated by considering the distance between the centre of mass of each individual cloud and the total domain-averaged centre of mass. In the case of a single object, L$_2$ is equal to 0, as the centre of mass in the domain is the same as the centre of mass of the individual object.

When the simulation and observations compare perfectly, the score of each of the components is 0. For the S and A score the values are all between $\pm 2$, where a value of -1 represents a factor of 3 underestimate of the simulation compared to the observations and +1 represents a factor of 3 overestimate of the simulation. For the L$_1$ and L$_2$ scores, values are between [0,1], with the worst possible score being 1 representing a distance equal to the maximum distance within the domain. Similar to Wernli et al. (2008), we will present the 3 components of this metric in a SAL-diagram (fig. 5d), where the horizontal axis represent the S score, the vertical axis the A score and the colour of each point represents the L score (sum of L$_1$ and L$_2$). The simulation and observations compare best when all the points are near the origin and have the dark purple colour.

## 3   Results

First we qualitatively discuss the spatial pattern of the SO$_2$ cloud and its dispersion across the NH during the first week after the eruption. We then discuss the FSS and the SAL-scores for the SO$_2$ cloud, followed by a discussion of the SO$_2$ mass burden evolution during the first 25 days after the eruption. A video of the volcanic SO$_2$ and SO$_4$ VCDs as simulated by NAME for the VolRes1.5 and StratProfile emission profiles can be found in the video supplements (de Leeuw, 2020).

### 3.1   Spatial pattern of the sulfur dioxide cloud

Qualitatively, the general structure of the SO$_2$ cloud simulated by the NAME VolRes1.5 and the StratProfile simulations compare well with the retrieved TROPOMI SO$_2$ VCDs during the first week after the eruption. On 23 June 2019 the TROPOMI retrievals show a split between the northern and southern branch of the SO$_2$ cloud, as seen in fig. 3. This observed SO$_2$ cloud structure was strongly influenced by a low-pressure cyclone approximately 1500 km to the east of the volcano. As a result of the low-pressure system, the volcanic cloud within the troposphere (below 11 km) moved predominantly in a south-eastward direction along the south-flank of the cyclone until it started to wrap around the centre on 23 June. For the cloud layers at higher altitudes within the stratosphere, the main wind direction was more zonal, resulting in the observed split in fig. 3. The VolRes1.5 and the StratProfile simulations show the same spatial pattern, but have different SO$_2$ VCDs within the cloud (see

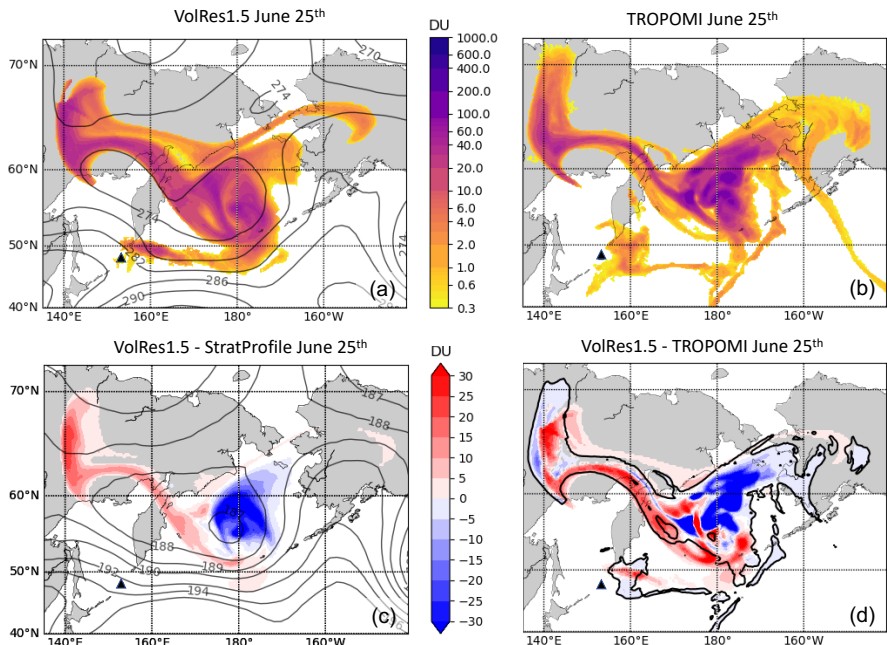

**Figure 6.** The spatial pattern of the volcanic cloud on 25 June 2019. The colours in panel a and b represent the $SO_2$ VCD for the VolRes1.5 simulation and TROPOMI retrievals respectively. The black contours in panel a) show the pressure at 10 km asl. Panel c) shows the difference between the VolRes1.5 and the StratProfile simulations and the black contours show the pressure at 12 km asl. The negative (blue) values indicate the part of the cloud within the stratosphere, while the positive (red) values highlight the cloud within the UT. Panel d) shows the difference between panels a and b, where the contour shows the 1 DU contour for the $SO_2$ cloud retrieved by TROPOMI in panel b.

e.g. fig. 3d). The StratProfile (fig. 3c) simulation shows a better agreement with the TROPOMI $SO_2$ VCD values for this day, which is expected based on its derivation.

By 25 June 2019, a large part of the cloud moves in a north-western direction, spreading over the Asian continent as seen in fig. 6a and fig. 6b for both the VolRes1.5 simulation and TROPOMI. Due to the variation in emission heights between the Vol-Res1.5 and the StratProfile simulations (fig. 2), we can identify the parts of the cloud in the NAME simulations that are mainly within the troposphere and the stratosphere by comparing their differences. The results are shown in fig. 6c, which shows that the north-western part of the cloud is mainly within the troposphere, while the stratospheric parts of the cloud remain centred around the low-pressure system. Calculating the difference between the VolRes1.5 simulation and the TROPOMI retrievals in fig. 6d, we find that the pattern is very similar to fig. 6c. This shows that the VolRes1.5 simulation mainly overestimates the $SO_2$ mass of the cloud in the troposphere and underestimates the stratospheric part of the cloud.

On 27 June 2019, the $SO_2$ cloud starts to spread also at higher altitudes, leading to a complex spatial pattern as shown in fig. 7. While the large-scale structure of the cloud on 27 June has become much more complex, both the VolRes1.5 and the StratProfile simulations capture the general $SO_2$ VCD structure of the retrieved TROPOMI cloud well. Note that the small-

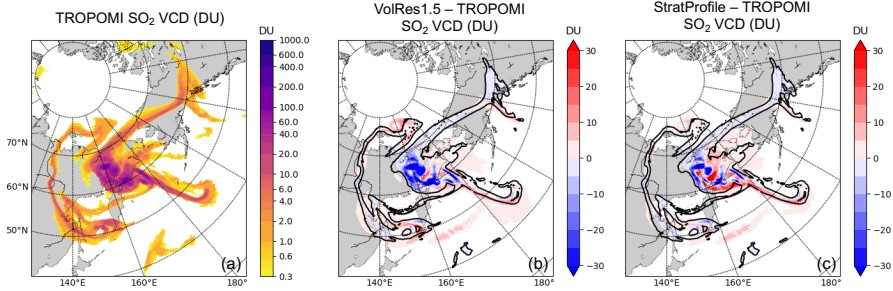

**Figure 7.** The spatial pattern of the volcanic $SO_2$ cloud as retrieved by the TROPOMI satellite for the 27 June 2019 (panel a). Panels b) and c) show the difference (in DU) with the VolRes1.5 and the StratProfile simulation respectively. Only $SO_2$ VCD values above 1 DU are shown. The contour shows the outline of the $SO_2$ cloud in TROPOMI for a $SO_2$ VCD of 1 DU.

scale eddies observed by TROPOMI in the centre of the cloud are not simulated by NAME as a result of the limited (spatial and temporal) resolution of the NWP input. Therefore, the small-scale variability cannot be captured by the model, but instead are parameterised by the diffusion parameters as a random perturbation on the wind field (see section 2.3.1). This results in the spreading of the $SO_2$ cloud with a smoother pattern in the NAME simulations without the high peak values. This also explains
the patchy variations shown in figs. 6b and 6c within the centre of the cloud. Averaging the $SO_2$ VCDs over the whole domain shown in fig. 7 (thereby removing the small-scale features from TROPOMI), the average $SO_2$ VCD values for the VolRes1.5 simulation are 20% lower than measured by TROPOMI. This is also evident from the dominant blue colours in fig. 7b. For the StratProfile simulation (fig. 7c) the domain-average mass is within 0.01 Tg of $SO_2$ of the TROPOMI $SO_2$ mass estimate (i.e. StratProfile $SO_2$ mass estimate is <1% lower than TROPOMI).

Finally, we also find a larger spread of the cloud in both the VolRes1.5 and StratProfile simulations as seen in figs. 7b and 7c by the red values outside the 1 DU TROPOMI contour. We only included the values > 1 DU in this plot for clarity of the figure. When including lower values (0.3-1 DU), the overestimation of the spread of the cloud in NAME is even larger (not shown here).

### 3.2   Fractional Skill Score (FSS)

The VolRes1.5 and the StratProfile simulations are generally able to capture the large-scale structure of the $SO_2$ cloud, but differences between the simulations and the satellite retrievals occur after 4-5 days of simulation (see for example fig. 6). To determine the timescales for which the simulations show skill compared to the TROPOMI retrievals, we calculate the FSS score for each individual overpass for a range of $SO_2$ VCD contours (ranging between 0.3 and 100 DU). The results for the smallest neighbourhood size N=1 (i.e. the NAME output gridbox size 0.2°latitude and 0.4°longitude) are shown in fig. 8 for
(a) the VolRes1.5 and (b) the StratProfile simulation.

The VolRes1.5 simulation is able to capture the overall outline of the cloud well for this period, but is struggling to simulate the peak $SO_2$ VCD values within the retrieved TROPOMI $SO_2$ cloud. Focussing on the $SO_2$ VCD > 1 DU points in fig. 8a,

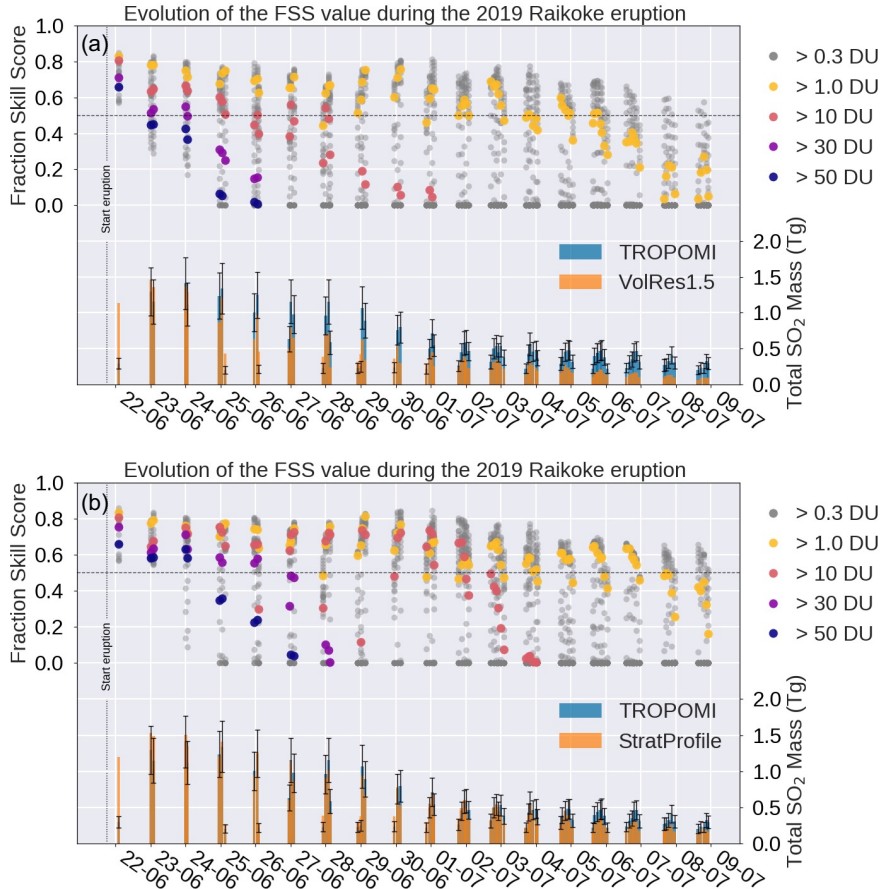

**Figure 8.** Time evolution of the FSS (N=1) and the $SO_2$ mass for each individual overpass of the TROPOMI satellite over the Raikoke cloud for a) the VolRes1.5 simulation and b) StratProfile simulation. Each annotated date represents 00UTC. Grey dots represent all concentrations between 0.3 and 100 DU, with the highest skill score for the lowest concentrations. The horizontal dashed line shows a value of FSS=0.5, which is the cut-off value for determining the skill of the simulations.

the simulation has skill (FSS> 0.5) for up to 12.5 days after the eruption onset. This shows that the simulation is capturing the overall dispersion of the cloud well, as it is able to distinguish between areas with and without any $SO_2$ across the NH. For $SO_2$ VCDs greater than 30 DU, which correspond to small-scale features within the volcanic cloud, the simulation has no significant skill beyond 2.5 days after the start of the eruption. This agrees with the fact that the VolRes1.5 simulation was not able to capture the peak values on the 25 June 2019 observed by TROPOMI as shown in fig. 6.

The FSS values for the StratProfile simulation (fig. 8b) reveal that this simulation performs better than VolRes1.5 and has skill on a longer timescale for all of the $SO_2$ VCDs. For the lower $SO_2$ VCDs (<1 DU), the StratProfile simulation remains skilful 2 days longer than the VolRes1.5 simulation (12.5 days versus 14.5 days). For the $SO_2$ VCDs above 30 DU, the FSS



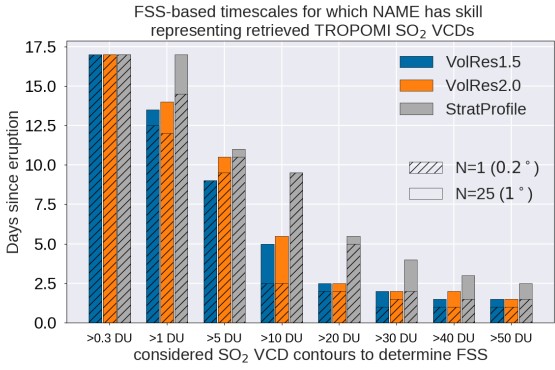

**Figure 9.** The timescales for which the NAME model shows skill based on the FSS, when compared with TROPOMI retrievals. The results are shown for the three simulations as discussed in section 2.3.4 and for two different neighbourhood sizes (value in brackets represent the corresponding resolution). The FSS metric is calculated only for the first 17 days of the simulation (up to 10 July), as the $SO_2$ VCD values become too small after to give a good estimate of the FSS from the TROPOMI measurements.

skill timescale has doubled compared to the VolRes1.5 simulation, showing again the importance of the emission profile on the skill of the simulation.

The timescales for which the NAME simulations show skill (compared to the TROPOMI retrievals) in terms of FSS are shown in fig. 9 and supplementary table A1. Independent of the neighbourhood size, the StratProfile simulation has the highest
skill for all $SO_2$ VCDs. Figure 9 shows that the StratProfile simulation is skilful on timescales twice as long for $SO_2$ VCD values above 10 DU compared to the VolRes1.5 and VolRes2.0 simulations. Interestingly the change in neighbourhood size (i.e. averaging region) has only a limited impact on the skill timescales for low $SO_2$ VCDs (below 5 DU). This shows that all of the simulations are able to capture the horizontal extent of the $SO_2$ cloud well on spatial scales similar to our smallest output grid used ($0.2° \times 0.4°$) and on timescales of 2-3 weeks after the start of the eruption.

The reduction in FSS scores for high $SO_2$ VCDs is influenced by two factors: 1) does the simulation capture high $SO_2$ VCDs? and if so 2) is the location of the high VCD features in the $SO_2$ cloud (see e.g. fig. 7c) correct? Due to the dispersion of the $SO_2$ cloud with time, we expect a decrease in the FSS values in time for the higher $SO_2$ VCDs as these concentrations are not present anymore in either the TROPOMI retrievals nor the NAME simulations (resulting in FSS=0). The skill timescales are therefore expected to reduce as the $SO_2$ VCDs increase. The StratProfile simulation contains higher $SO_2$ VCDs throughout
the simulation period compared to the VolRes1.5 and VolRes2.0 simulations, resulting in higher FSS values and longer relative skill.

For high $SO_2$ VCDs, the FSS metric depends more on the used neighbourhood sizes as the corresponding $SO_2$ cloud features get smaller. Using a larger neighbourhood size compares the presence of small-scale features over a larger region, reducing the impact of any misplacement, and results in a higher FSS. The $SO_2$ VCD values at which model skill increases for different
neighbourhood sizes is therefore linked to a displacement error. In fig. 9, a doubling in skill timescales is found for the larger neighbourhood size (hashed versus non-hashed) at $SO_2$ VCDs >10 DU for the VolRes1.5 and VolRes2.0 simulations, while

for the StratProfile similar differences are evident for $SO_2$ VCDs above 30 DU. This shows that the VolRes1.5 and VolRes2.0 simulations are able to represent observed small-scale features within the $SO_2$ cloud for VCDs up to 10 DU at timescales less than 5 days. For $SO_2$ VCDs >20 DU these two simulations gain no additional skill with an increased neighbourhood size and show a strong reduction in skill timescales. This indicates that the high $SO_2$ VCDs (>10 DU) observed by TROPOMI are not simulated anywhere in the $SO_2$ cloud at timescales longer than 5 days. For the StratProfile simulation, features with $SO_2$ VCDs above 30 DU are still present up to 4 days. However, these features are slightly displaced, as evident from the increase in the FSS skill timescales from increasing the neighbourhood size from 0.2° to 1°.

On timescales longer than 5 days, all the NAME simulations show a strong diffusion in the $SO_2$ cloud (related to the diffusion parameterisations). As a result none are capturing the high $SO_2$ VCDs retrieved by TROPOMI, which reduces the FSS for these high values quickly to 0. This shows that high $SO_2$ VCDs within the $SO_2$ cloud are only skilfully simulated on timescales less than 5 days.

### 3.3 The SAL-score

Figure 10 shows the SAL-scores for all the individual TROPOMI overpasses and the daily average values for four different NAME simulations. This comparison shows the strength of the SAL-diagram to determine what aspects of the $SO_2$ cloud are captured well by the NAME simulations and where the simulations are struggling to match the TROPOMI retrievals. At the start of the eruption, all simulations are in the top half of the diagram (positive A score). This indicates that the simulations have a larger total $SO_2$ mass than the TROPOMI retrievals during the first days after the eruption. This can be partly explained by the presence of ash interfering with the TROPOMI retrievals (see section 3.4). Furthermore the S-values are close to 0, indicating that the shape of the cloud is captured well within the simulations. The low L values throughout all the simulation indicate that the location of the $SO_2$ clouds is well captured.

All four simulations shown in fig. 10 show a tendency of increasing positive S score values with time, with a strong increase on 27 June, which is 5 days after the start of the eruption. An increase in the S score represents an $SO_2$ cloud which is more widespread (platykurtic) in the simulations compared to the $SO_2$ VCDs obtained from TROPOMI. The largest changes in the S score around 5 days into the simulation are consistent with our FSS analysis where we identified that this is also the time where the VolRes1.5 and the StratProfile simulations are losing the skill to represent high $SO_2$ VCDs retrieved by TROPOMI (i.e. not capturing the peak values in the cloud).

Focussing on the VolRes1.5 and the VolRes2.0 simulations (figs. 10a and 10b), both SAL-diagrams show a similar pattern (moving from the top left towards the bottom right in the diagram). Due to the total emitted $SO_2$ mass being greater in the VolRes2.0 simulation, fig. 10b shows a more positive A score during the first 4 days after the eruption (22-26 June) than the VolRes1.5 simulation as the former overestimates the total $SO_2$ mass retrieved by TROPOMI. After 26 June, the A score for VolRes2.0 remains close to the A=0 line in fig. 10b, showing that the total $SO_2$ mass compares better with TROPOMI for the VolRes2.0 simulation than the VolRes1.5 simulation after 5 days. For the StratProfile simulation (panel 10c), the comparison with TROPOMI is better throughout the entire simulation than for the VolRes1.5 and the VolRes2.0 simulations, as is evident by the low A score as well as the low L score. The comparison between the StratProfile simulation and TROPOMI for each

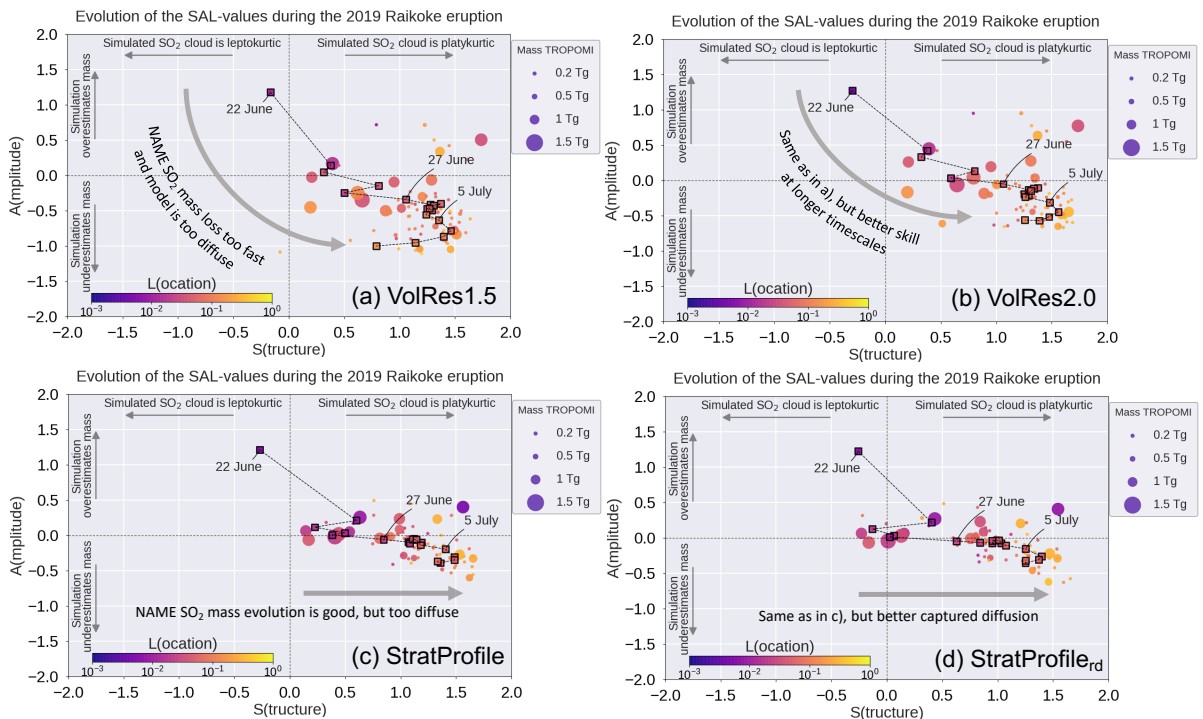

**Figure 10.** Time evolution of the SAL-values for 4 different NAME simulations: a) VolRes1.5, b) VolRes2.0 c) StratProfile and d) StratProfile_rd. The black dashed line shows the daily average evolution of the S and A parameters between 22 June and 10 July, while the coloured squares show the daily average values of L. The coloured dots represent the SAL-values for each individual TROPOMI overpass as shown in figure 2. To obtain the A parameter, we have excluded the $SO_2$ VCDs of TROPOMI below 0.3 DU to reduce noise but included all mass for the NAME simulations (see sections 2.3.2).

individual TROPOMI overpass (i.e. each dot in fig. 10c) is close to the A=0 line in the diagram, showing that NAME is able to capture the total $SO_2$ mass very well. Also a lower L score (darker colour of the squares and circles) indicates that the model is capturing the location of the cloud more accurately than both the VolRes simulations. These results are consistent with the results shown in figs. 6-9.

Reducing the horizontal diffusion parameter by 75% ($\boldsymbol{K_{meso}}$, see table 2) in the StratProfile_rd simulation (panel 10d), reveals a relative decrease in the S score during the first week (e.g. S=0.5 for StratProfile vs S=0.1 for StratProfile_rd on 25 June) and no change in the A and L scores when compared to fig. 10c. This behaviour is expected, as a decreased diffusion will not alter the total mass (A score) nor the centre of mass of the individual $SO_2$ clouds (L score), but it will result in a more concentrated $SO_2$ clouds (reduction of S). As a result, the StratProfile_rd simulation shows the best comparison with the TROPOMI retrievals for the first 4 days of the eruption (up to 26 June). After that the S score quickly increases for

all the simulations independent of the diffusion parameterisation, as diffusion related to uncertainties from the large-scale meteorological conditions (i.e. synoptic scale uncertainties) starts to dominate the signal.

### 3.4 Sulfur dioxide mass burden

Figure 11a shows the $SO_2$ mass burden evolution calculated from the TROPOMI satellite retrievals and three NAME simu-
lations (VolRes1.5, VolRes2.0 and the StratProfile) for the 25 days between the start of the eruption and 15 July 2019. The best comparison is obtained using the StratProfile simulation, which captures both the peak value and the long-term evolution remarkably well and falls well within the uncertainty range of the TROPOMI estimate. To obtain the mass burden, we have excluded the $SO_2$ VCDs of TROPOMI below 0.3 DU to reduce noise but included all mass for NAME simulations as discussed in the methods section. It is likely that TROPOMI underestimates the $SO_2$ VCDs and thus $SO_2$ mass during the initial phase of the eruption due to the presence of volcanic ash, which is supported by large values of AAI obtained from TROPOMI during the first 2 days after the eruption (see grey bars fig. 11a and also section 2.1).

Consistent with figs. 3, 6 and 7, the VolRes1.5 simulation captures the peak in total $SO_2$ mass within the uncertainty of the TROPOMI estimate, but underestimates the TROPOMI $SO_2$ mass between 27 June 2019 and 15 July 2019 by 0.3 Tg on average. Based on the data shown in figure 11, we calculated an e-folding time of $\approx$9 days for the VolRes1.5 simulation and an e-folding time of $\approx$21 days for TROPOMI during the first 6 days after the eruption (23-27 June 2019), showing that the VolRes1.5 simulation loses $SO_2$ mass at a much faster rate than that calculated based on TROPOMI. As a result, this leads to an underestimation of 25% (0.33 Tg) of the VolRes1.5 simulation compared to TROPOMI on 28 June. From 27 June the loss rate for both TROPOMI and VolRes1.5 is similar, with an e-folding time of $\approx$14-15 days, which is within the range reported in the literature for extratropical summer eruptions of similar magnitude (e.g. e-folding time of 9-18 days for Kasatochi 2008 and 11-14 days for Sarychev 2009) (Karagulian et al., 2010; Krotkov et al., 2010; Haywood et al., 2010; Jégou et al., 2013; Höpfner et al., 2015; Carn et al., 2016).

After 27 June 2019, the total $SO_2$ mass burden evolution from TROPOMI is best captured by the VolRes2.0 and the Strat-Profile simulations, related to a larger amount of mass emitted into the stratosphere. From the total $SO_2$ mass emitted into the stratosphere calculated in table 2, we see that both the VolRes2.0 and the StratProfile respectively emit 0.2 and 0.45 Tg more $SO_2$ mass into the stratosphere than the VolRes1.5 simulation. For the VolRes2.0 simulation the overall evolution of the $SO_2$ mass profile is similar to that obtained from the VolRes1.5 simulation (e-folding time of $\approx$9 day during the first week, e-folding time of 14-15 days afterwards). However, due to the increased total $SO_2$ emissions (0.5 Tg more than VolRes1.5), the $SO_2$ mass evolution of VolRes2.0 also overestimates the TROPOMI peak mass by more than 0.5 Tg on 23 June. Given that TROPOMI is used as our baseline metric for initialising our StratProfile simulations, it is not surprising that the best comparison with TROPOMI during the start of the eruption is obtained for the StratProfile simulations. However, on longer timescales the influence of other factors (e.g. simulated wind field, radiative heating, mixing) on the dispersion of the $SO_2$ cloud means that the model simulation could easily diverge from the observations. That the StratProfile continues to best match the TROPOMI data gives confidence that NAME captures the main processes needed to represent the $SO_2$ dispersion well.

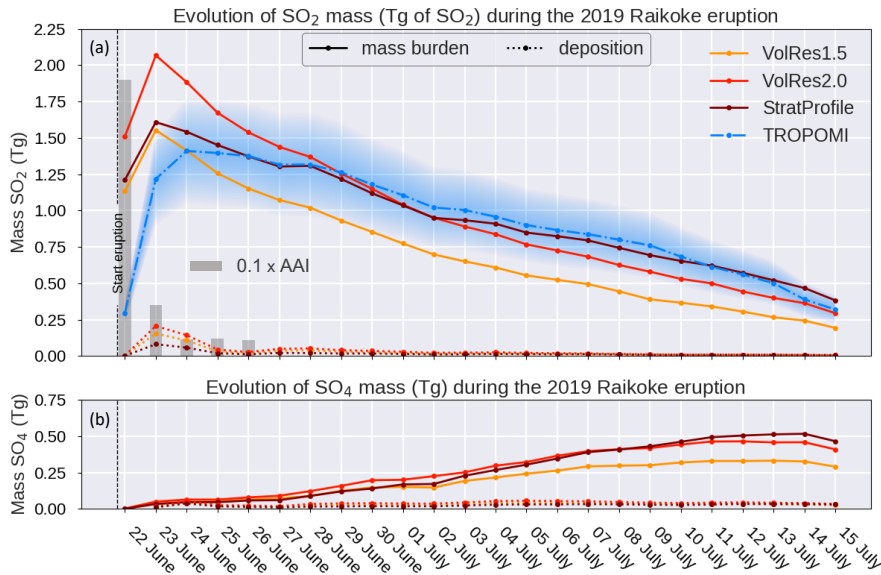

**Figure 11.** The daily evolution of a) the total $SO_2$ mass (Tg of $SO_2$) and b) the total $SO_4$ mass (Tg of $SO_4$) for the 2019 Raikoke eruption for different ESP in NAME. We have included the TROPOMI $SO_2$ mass estimate (blue dashed line) as well as the evolution of 3 NAME runs: VolRes1.5, VolRes2.0 and StratProfile (see fig. 2). The dotted lines in the figures show the corresponding daily deposition of $SO_2$ and $SO_4$ from the simulations. The peak values in the NAME $SO_2$ mass distribution are slightly higher than the mentioned total emission values in table 2, which is the result from applying the 15 km AKs to the dispersion model data. The total $SO_2$ mass burden for TROPOMI is calculated using all locations where the vertical column densities are above 0.3 DU, while for the NAME simulations we include all mass. The blue shading represents the standard error estimate for the TROPOMI product. The grey bars show the TROPOMI estimated $0.1 \times$ max(AAI) value inside the volcanic cloud for the first 5 days after the eruption. The high AAI values during the first 48 hours indicate high concentrations of ash, thereby affecting the TROPOMI $SO_2$ retrievals during this period.

A possible cause for the strong reduction in total $SO_2$ mass during the first week for the VolRes1.5 and VolRes2.0 simulations might be too strong a conversion of $SO_2$ into sulfate aerosols during the start of the simulation. To test this hypothesis, we also calculated the mass evolution of $SO_4$ in NAME, which are shown in fig. 11b. From this we can conclude that the chemical conversion into $SO_4$ is realistic within the NAME simulations. The daily rate of production of $SO_4$ is small (less than 0.03 Tg/day), which is a factor of 3 lower than the average daily decrease in $SO_2$ mass in the VolRes1.5 and VolRes2.0 simulations during the first week (0.1 Tg/day).

The daily total $SO_2$ mass deposition from the NAME simulations is shown by the dotted lines in fig. 11a. For 23 and 24 June, the total daily wet deposition dominates the removal of $SO_2$ from the atmosphere, as it is responsible for 89-90% of the $SO_2$ mass reduction for the VolRes1.5, the VolRes2.0 and the StratProfile simulations. Atmospheric conditions during the first week of the eruption can explain this relatively large contribution from wet deposition. During the first week of the eruption, the $SO_2$ cloud is moving within a region of moist air in the warm conveyor belt on the southern edge of the cyclone (see

fig. 3). This favours the chemical conversion of $SO_2$ into $SO_4$ through aqueous-phase chemistry and also the removal of $SO_2$ through wet deposition, resulting in the peak deposition values in fig. 11a. The cyclone is mainly a tropospheric phenomenon and as a result wet deposition occurs mostly in the tropospheric part of the $SO_2$ cloud. As the VolRes2.0 simulation emits the largest amount of $SO_2$ into the troposphere (see fig. 2), this also explains the highest removal rate (red dotted line fig. 11a peaks at 12% of the NH-mean daily $SO_2$ mass burden on 23 June) and also the highest conversion rate during the first week

of the simulation (evident from the largest $SO_4$ mass burden in fig. 11b in this period). The wet deposition is lowest for the StratProfile simulation during 23 and 24 June (peaks at 4-5% of the NH-mean daily $SO_2$ mass burden on 23 June), as less mass is emitted into the troposphere for this profile.

The results from fig. 11 show that there is a high sensitivity of the mass burden evolution in NAME to the vertical emission profile used for this particular eruption, which straddled the tropopause. Due to different atmospheric conditions within the

troposphere and stratosphere, the $SO_2$ mass burden evolution is different within the two layers, resulting in significant differences in the total $SO_2$ mass burden evolution in our simulations. The average e-folding time of $SO_2$ in the UT is $\approx 10$ days (e.g., Krotkov et al., 2010; Carn et al., 2016), which is consistent with the e-folding time simulated during the first days of the VolRes1.5 and VolRes2.0 simulations. However, the longer e-folding time obtained for TROPOMI for the first 10 days suggests that the bulk of the $SO_2$ mass was not emitted into the UT.

After 10 days a large fraction of the tropospheric $SO_2$ mass is removed from the atmosphere through wet deposition or converted into $SO_4$ and the remaining signal in fig. 11 is dominated by the stratospheric component of the cloud. This part of the cloud is much less affected by the cyclone and the stratosphere contains much less moisture. Therefore $SO_2$ is removed at a much lower rate ($SO_2$ deposition is <1%) and is mainly converted through the gas-phase reaction with OH, resulting in the longer e-folding time of approximately 14-15 days. The similarity in e-folding time obtained from TROPOMI and all the

simulations between 27 June and 15 July suggests that the chemistry scheme in NAME is realistic.

Overall, the total $SO_2$ mass burden obtained using the StratProfile emission profiles (both StratProfile and $StratProfile_{rd}$ give the same mass evolution) compares best with TROPOMI. Based on this comparison, we estimate that the 2019 eruption of Raikoke emitted approximately 0.9-1.1 Tg of $SO_2$ into the lower stratosphere (11-18 km asl). With a maximum $SO_2$ mass burden of 1.5-1.6 Tg in the atmosphere, it follows that approximately 0.4-0.7 Tg was emitted into the UT (8-11 km asl).

**4  Discussion**

Our study shows that the NAME simulations compare very well with the TROPOMI satellite retrievals of the $SO_2$ cloud during the first three weeks after the 2019 Raikoke eruption. Despite the increasing complexity of the $SO_2$ cloud's horizontal structure over time, all our simulations are able to capture the outermost extent of the cloud within an accuracy of approximately 0.4°(50 km) during the first two weeks of the simulations and for up to 25 days with an accuracy of approximately 1°(100 km) (see

figs. 8 and 9 and supplementary table A1). While simulated $SO_2$ concentrations within the cloud are strongly dependent on the ESPs, the general dispersion patterns of the $SO_2$ cloud are captured very well in both the troposphere and the stratosphere for all the NAME simulations performed (see figs. 3, 6 and 7). Combining this information with the comparison of the vertical

profile from IASI (fig. 4) and the included representation of the sulfur chemistry in the NAME simulations (fig. 11) gives us confidence that the NAME model is able to simulate the 3D structure of the volcanic $SO_2$ cloud (and consequently the $SO_4$ cloud) for the 2019 Raikoke eruption.

While the NAME model was not developed specifically to simulate stratospheric volcanic $SO_2$ clouds, our results show that the model is suitable to be used by VAACs to issue forecasts on the evolution of volcanic $SO_2$ clouds in the upper troposphere/lower stratosphere. Currently after a volcanic eruption, VAACs provide information on the areas in the atmosphere where volcanic ash is forecasted up to 18 hours into the future. In the case of the Raikoke 2019 eruption, we have shown that a similar approach to produce a forecast for the presence of a $SO_2$ cloud would have been accurate on this and even longer timescales. However, future $SO_2$ cloud forecasts will more likely be based on designated $SO_2$ concentration thresholds. From our simulations we found that NAME is able to capture the horizontal extent of the 1 DU $SO_2$ VCD contour on a spatial resolution of $0.2° \times 0.4°$during the first 17 days after the eruption. Assuming that the obtained $SO_2$ VCD values are from a cloud at 12 km altitude with a thickness of 2 km (e.g. estimated from fig. 4), 1 DU would correspond to an average $SO_2$ concentration of 0.02 ppm within this cloud. As reference, based on sulfur dioxide Acute Exposure Guideline Levels (AEGL) (National Research Council Committee, 2010), an extended exposure (> 10 minutes) to $SO_2$ concentrations of 0.2 ppm (lowest AEGL) can lead to some respiratory irritation, while concentrations above 0.75 ppm can lead to long-lasting adverse health effects. For our example, the lowest AEGL threshold would therefore correspond to $SO_2$ VCDs above 10 DU. We find that NAME is capable of capturing the spatial distribution of the features within the $SO_2$ cloud where the $SO_2$ VCDs are larger than 10 DU on the order of 7-10 days (see fig. 9).

Our work highlights that accurate information on ESPs is key when comparing model simulations to satellite retrievals. In reality it can be difficult to obtain this information, especially in near real-time. For example, observations show that the Raikoke eruption was actually characterised by a series of explosive eruptions that emitted $SO_2$ at varying heights (Crafford and Venzke, 2019; Hedelt et al., 2019; Muser et al., 2020; Kloss et al., 2021). Furthermore a study by Prata et al. (2020) suggests that activity continued until 10 UTC on 22 June rather than 03 UTC as reported by the VolRes team. These examples indicate how hard it is to obtain a 'correct' time-dependent emission profile for the eruption in near real-time. Instead, the profiles used in this study try to capture the 'time averaged' emission profiles of these 'pulses' during the eruption. To test the importance of the emission duration, we conducted some additional simulations where we increased the duration of the eruption to 10 UTC on 22 June. The results of these new simulations (not shown) were very similar to the results presented in this paper, with the main difference that the structure of the cloud is slightly more diffuse.

For all the simulations we find that small changes in the vertical emission profile lead to a large change in the amount of mass emitted into the stratosphere, due to the emissions spanning the tropopause. In our analysis the mass-averaged emission height in the VolRes and StratProfile emission profiles differs by only 1 km (10.5 km in VolRes versus 11.5 km in StratProfile). But in the StratProfile emission profile we emit 69% (1.09 Tg) of the total $SO_2$ mass above the tropopause (as defined by the MetUM NWP), compared to 43% (0.64 Tg) in the VolRes1.5 simulation.

NAME does not account for any radiative lofting effect and thus an emission profile derived during the first hours after the eruption could lead to $SO_2$ at the wrong altitudes. Muser et al. (2020) shows that for the Raikoke 2019 multi-phase plume, the

radiative lofting effect for ash is in the order of 2-3 km during the first days after the eruption. While the precise impact of such a lofting effect on the $SO_2$ cloud is difficult to quantify (owing to lack of knowledge of the details of the relative vertical position on the $SO_2$ and the ash, and the lack of knowledge of the mixing state between the resulting sulfate aerosol and ash), it could help to explain the differences seen between the emission profiles used in this study. For the VolRes1.5 emission profile, which was derived during the first hours after the eruption, the radiative lofting effect is still limited. Instead for the StratProfile, which is derived 30 hours after the eruption onset, the radiative lofting has very likely impacted the $SO_2$ cloud vertical structure as Muser et al. (2020) shows that most of the lofting took place during the first 36 hours after the Raikoke eruption. Therefore, we get a better comparison between the NAME simulations and observations by determining an 'effective' emission profile from 1-2 days after the initial emission time to account for potential radiative lofting effects. Our study results show that the potential impact of radiative lofting should be considered by VAACs when producing forecasts for multi-phase plumes near the tropopause.

Interestingly while the fractional split of the mass between the stratosphere and troposphere is important, the detailed vertical distribution of $SO_2$ within each of these two layers has only a minor effect on model performance. To test the sensitivity of the model results to the details of the vertical distribution of the emissions within the troposphere and stratosphere, we performed a simulation where we emitted 1 Tg evenly between 11-14 km and 0.5 Tg evenly between 9-11 km (not shown). The general conclusions were the same as shown for the StratProfile, illustrating that, for the 2019 Raikoke eruption, it is key to establish the fraction of the $SO_2$ mass that was emitted into the stratosphere.

Also an error in the definition of the tropopause height within the MetUM (or any other NWP) model can have a large influence on the skill of the dispersion simulations. If, for example, the MetUM model-estimated tropopause height is 2 km above the actual observed tropopause height, this would lead to a wrong placement of the majority of the $SO_2$ mass in the troposphere in our VolRes1.5 simulation. Using a different NWP model with a lower tropopause height in the NAME simulations would result in a higher fraction of the $SO_2$ mass of the VolRes1.5 profile being emitted into the stratosphere and potentially give a better comparison with TROPOMI than the StratProfile estimate (which would overestimate the stratospheric $SO_2$ mass in that case). However, we found the average tropopause height of $11.2\pm0.7$ km diagnosed in the model at the eruption site is in good agreement with the observed tropopause height of $10.5\pm0.7$ km from a nearby radiosonde location (see fig. 2), which gives us confidence that our StratProfile emission estimates for the stratosphere and troposphere are suitable and that the NAME results are not strongly biased by a wrong tropopause height within the NWP fields.

While we have investigated the impact of changing the vertical $SO_2$ emission profile (see fig. 2), we have not investigated the effect of any uncertainty related to the atmospheric conditions in the NWP wind fields used as input for the NAME simulation. A study by Dacre and Harvey (2018) shows that the impact of the atmospheric conditions on the NAME simulations can be large, especially in conditions of large horizontal flow separation in the atmosphere. The specific atmospheric conditions for this particular eruption (i.e. the low-pressure system east of the eruption site) shows isobars that are parallel to each other during the first days after the eruption in the region of the cloud (fig. 3). Therefore, it is expected that the impact of flow separation is limited during the initial stages of the cloud evolution. After several days, the flow separation becomes more pronounced in various regions in the domain (see fig. 6). This effect is also reflected in the decrease in the S score value of the SAL diagram

shown in fig. 10, as trajectories start diverging in these regions of enhanced flow separation enhancing dispersion in the model. To better understand the impact of the NWP wind field variability on our results presented here, the analysis would need to be repeated using an ensemble of NWP wind field forecasts as input for the NAME simulations.

Our study of the 2019 Raikoke eruption demonstrates the strength of the SAL-diagram for performing a comparison of model simulations with satellite observations and can help to determine potential issues. In this particular case, the NAME model simulations generally tend to show increasing positive S score values for all the simulations. Partly this is related to the diffusion parameterisation in the model, which smooths the signal (i.e. random perturbation) of the small-scale eddies within the cloud that are not present in the input wind field (see section 2.3.1). As uncertainties from the meteorological conditions used as input for the simulations gradually accumulate, this leads to a larger spread in the $SO_2$ cloud over time than is observed in reality. This gives the tendency for the model to be more diffuse on longer timescales, as revealed by the S score increase in figs. 10a-c.

We find that the key reason for the increasing S score values on shorter timescales (i.e. 1-5 days) is related to the horizontal diffusion parameter $K_{meso}$ used for the model simulations. NAME v8.1 uses a single value for the diffusion coefficient within the free atmosphere (see table 1). From literature it is known that mixing in the atmosphere can be highly variable and seasonally dependent (e.g., Haynes and Shuckburgh, 2000; Allen and Nakamura, 2001; Legras et al., 2005; Abalos et al., 2016). Furthermore, the diffusion parameter values currently used in NAME have been determined using observational datasets near the surface (Webster et al., 2018). It is therefore possible that the mesoscale diffusion values used in the model might be unsuitable for the higher levels in the atmosphere, especially in the stratosphere and thereby cause too much diffusion from the start of the eruption. To test this hypothesis, we investigated several simulations with a smaller horizontal diffusion coefficient (see table 2), where we reduce the mesoscale diffusion value $K_{meso}$ by 75% for the whole atmosphere above the boundary layer. The resulting values for the FSS-score and SAL-diagram for the StratProfile$_{rd}$ simulation are shown in fig. 10d and table A1 and indicate that the simulations are better able to capture the structure (i.e. peak values and horizontal extent) of the cloud during the first 5 days of the simulations. It is currently impossible to determine a more precise value for the diffusion parameters, due to the lack of case studies and limited available observations for these high altitudes. Part of ongoing work is to investigate the impact of a new space and time-varying free-atmospheric turbulence scheme that is included in the latest version of NAME (Dacre et al., 2015), which was not available for the simulations presented in this paper.

A previous study by Harvey et al. (2018) used a multi-level emulation approach to better understand the influence of model parameters on the accuracy of NAME output for volcanic ash concentration during the 2010 Eyjafjallajökull eruption. Their study showed a limited impact of the mesoscale diffusion parameter $K_{meso}$ on their ash simulations for the 2010 Eyjafjalla-jökull eruption. This limited effect of $K_{meso}$ is partly explained by the lower resolution ($40 \times 40$ km) of their model output. This leads to an averaging of small-scale eddies and a reduced impact of the diffusion parameterisation, something we also observed in our study when using a larger neighbourhood size N in our calculations of the FSS score (see fig. 10 and table A1). Furthermore, the 2010 Eyjafjallajökull emissons were at lower altitude than for the Raikoke eruption and Harvey et al. (2018) only investigated the diffusion of the ash clouds, so a larger $K_{meso}$ might be more appropriate. However, the reduced $K_{meso}$ values in our StratProfile$_{rd}$ simulations are within the range of realistic values for the free atmosphere (see table 1 in

Harvey et al. (2018)) and so this motivates more research to investigate the potential improvement of the model by having a more detailed representation of the mesoscale diffusion in the model at higher altitudes.

The results from our work are for a single case study, as 2019 Raikoke is the first eruption of this magnitude that has been observed by the high-spatial resolution TROPOMI instrument. Nonetheless, our work shows the large potential of using TROPOMI $SO_2$ retrievals (in combination with the correct AKs) to identify and rectify issues in dispersion modelling efforts of volcanic eruptions. Comparing high-resolution satellite measurements and dispersion model simulations is a valuable exercise that can help improve both the volcanic dispersion modelling tools and the satellite retrievals of volcanic plumes. By combining

the information obtained from NAME, TROPOMI and IASI for Raikoke 2019, we are able to give a more detailed picture of the eruption source parameters and the dispersion of the volcanic $SO_2$ cloud than ever before. Even though we only considered this one case study, it becomes clear that improvements in ADMs simulating volcanic eruptions can be expected when more volcanic eruptions are investigated using this or similar frameworks.

While this study has focussed almost entirely on representing the evolution of the gas-phase sulfur in the form of $SO_2$, it

is acknowledged that this is only part of the story. The Part 2 companion paper (Osborne et al., 2021) focusses on assessing the fidelity of the NAME model in representing the resulting sulfate aerosol together with volcanic ash and any confounding effects from biomass burning aerosol that were emitted into the stratosphere from an unusually strong pyrocumulus event in continental north America.

## 5   Conclusions

Volcanic eruptions can pose a large threat to society and in particular the aviation industry. In this study we simulated the 2019 Raikoke eruption using the Met Office's Numerical Atmospheric-dispersion Modelling Environment (NAME). The 21-22 June 2019 Raikoke eruption emitted $1.5 \pm 0.2$ Tg of $SO_2$ into the UT/LS. We evaluated the skill and limitations of NAME to simulate the dispersion of the resulting volcanic $SO_2$ cloud by comparing our simulations to high-spatial resolution satellite measurements from the Tropospheric Monitoring Instrument (TROPOMI). Based on our analysis we conclude that:

– NAME accurately simulates the observed location and horizontal extent of the $SO_2$ cloud during the first 2-3 weeks after the eruption (figs. 6 and 7). Based on the Fractional Skill Score (FFS), we find that our simulations have skill out to 12-17 days when considering the 1 DU $SO_2$ VCD contour (fig. 10 and table A1). For $SO_2$ VCDs larger than 20 DU, the model performs less well and has skill on the order of 2-4 days only (fig. 9).

– Based on both the FFS score and Structure, Amplitude and Location (SAL) metric (fig. 10), we find that the model-

simulated $SO_2$ cloud in NAME is more diffuse than in the TROPOMI measurements, in particular for $SO_2$ VCDs exceeding 20 DU. The diffusion parameterisation in NAME, which is developed for the lower free troposphere, results in too much horizontal spread in the lower stratosphere and therefore leads to a fast reduction of $SO_2$ mass concentrations in the densest parts of the eruption cloud right from the start of the eruption. This is reflected by the high positive S score of the SAL diagram shown in fig. 10 when compared to TROPOMI. Reducing the horizontal diffusion parameters in

NAME results in a better agreement during the first five days of the simulation (fig. 10d), but has no significant impact on timescales longer than that. We therefore suggest that the horizontal diffusion parameters currently used in NAME are potentially too large in the UT and LS and different values should be considered when investigating dispersion processes near the tropopause and in the stratosphere. Our reduced horizontal diffusion parameter values are within the range of values found in the literature for LS mixing (e.g., Balluch and Haynes, 1997; Waugh et al., 1997). However, we are not able to determine exact values for the diffusion parameters on the basis of this single volcanic eruption case study with one NWP representation and one set of detailed observations. In the future, as more eruption case studies become available, there is a potential to constrain the values of these diffusion parameters in order to better represent the diffusion of upper tropospheric and stratosphere volcanic $SO_2$ clouds in NAME and other ADMs.

– For the 2019 Raikoke eruption, we find that the skill of the model strongly depends on the eruption source parameter used for the simulation. Using the Volcanic Response team profile (VolRes1.5 fig. 2), we find that NAME removes too much $SO_2$ mass from the atmosphere during the first week of the simulations (fig. 11), resulting in a shorter e-folding time in the simulation than estimated based on TROPOMI data (e-folding time of 9 days for VolRes1.5 versus 21 days in TROPOMI between 23-27 June). A large fraction of the tropospheric $SO_2$ mass is removed from the atmosphere in the Warm Conveyor Belt region of the cyclone east of the eruption site during the first week of the simulation. NAME performs better with the StratProfile emission profile, where a larger fraction of the total $SO_2$ mass is emitted into the stratosphere (1.09 Tg versus 0.64 Tg, see fig. 2 and table 2). When emitting 0.9-1.1 Tg of $SO_2$ into the LS and 0.4-0.7 Tg into the UT, we obtain the best agreement with TROPOMI, both in terms of the peak $SO_2$ mass burden and the e-folding time of $SO_2$ (e-folding time of 14-15 days).

Determining the vertical $SO_2$ emission profile for any volcanic eruption from observations is one of the most difficult and challenging tasks the community faces. Our analysis shows that determining the details of the vertical profiles is essential in particular for eruptions that only just straddle the stratosphere in order to accurately forecast the dispersion of volcanic $SO_2$. This study demonstrates that combining observational datasets with dispersion model estimates can be beneficial to obtain a more detailed estimate of the volcanic $SO_2$ emission profile. The attempts in this paper to give more details on the vertical emission profile are basic and more sophisticated near realtime estimates are currently being developed by the scientific community (e.g., Kristiansen et al., 2010; Boichu et al., 2013; Moxnes et al., 2014; Vira et al., 2017; Pardini et al., 2018). With the current efforts, we expect that estimates of volcanic $SO_2$ fluxes will become more detailed and will very likely lead to a significant improvement of model simulations of future eruptions.

Finally, the average cruise altitudes for long-haul aircraft (11.9-13.7 km) fall well within the altitudes where the volcanic $SO_2$ cloud for the 2019 Raikoke eruption was observed (see fig. 4), thus eruptions like the 2019 Raikoke eruption may pose a hazard to the aviation industry. Having reliable dispersion models to simulate volcanic clouds are crucial to better understand and mitigate their potential impacts. We have shown that the FSS and the SAL-score metrics are potentially very powerful tools when assessing the skill of the model simulations in comparison with satellite measurements. The FSS score gives insight into the timescales over which the model has skill and also shows at what resolution results are significant. The SAL-score

gives a more detailed overview on three different aspects of the cloud properties (shape, location and mass) and helps to identify what aspects of the eruption cloud are well represented in a model. Using the two metrics in tandem gives a good overview of the strengths and weaknesses of the simulation and helps to interpret the results of the forecast model in more detail. While we have applied the metrics to the NAME model and the TROPOMI retrievals, they can also be easily applied to any combination of dispersion models and spatial observations. It could therefore also be a useful tool to inter-compare skills of satellite observation products and/or multiple dispersion models.

*Code and data availability.* Code and simulation data used in this manuscript may be requested from the corresponding author and can be downloaded from https://doi.org/10.5281/zenodo.4729991 (de Leeuw, 2021). The TROPOMI satellite data can be downloaded from the ESA website (https://s5phub.copernicus.eu, last access: 9 December 2020) (Copernicus, 2020). I.T. and R.G. plan to archive the Oxford IASI $SO_2$ products for the Raikoke eruption; in the meantime these can made available on request from Isabelle Taylor (isabelle.taylor@physics.ox.ac.uk). Radiosonde data are available at http://weather.uwyo.edu/upperair/sounding.html. The NAME code is available under license from the Met Office.

*Video supplement.* Videos of the $SO_2$ and $SO_4$ NAME VCD simulations for the VolRes1.5 and the StratProfile emission profiles can be found at http://doi.org/10.5281/zenodo.3992052 (de Leeuw, 2020).

## Appendix A: Derivation of StratProfile vertical emission profile

To investigate the sensitivity of the results to the $SO_2$ mass fraction that is emitted into the stratosphere, we derive a new emission profile based on the TROPOMI $SO_2$ VCD on 23 June in combination with the 36-hour NAME VolRes1.5 simulation. Using fig. 3d, we scale the VolRes1.5 $SO_2$ VCD estimates at each longitude to represent the TROPOMI $SO_2$ VCD values along cross section I-II and using fig. 4 we simultaneously derive the corresponding cloud height for the VolRes1.5 vertical profile at each point along the cross section. The obtained scaling and altitude of the cloud at each longitude are then combined to apply a scaling to the VolRes1.5 emission profile (fig. 2). We only determine the scaling where the modelled cloud thickness is less than $4\,\text{km}$ (between 157°E and 176°E), resulting in a scaling factor for the emission profile levels between 9 and $18\,\text{km}$. The resulting StratProfile shown in fig. 2 emits 1.57 Tg of $SO_2$ mass, which is very similar to the mass emitted by VolRes1.5. However, this new profile emits a much higher fraction of the mass into the stratosphere (1.09 Tg versus 0.64 Tg, see table 2). The results for the StratProfile simulations on 23 June are presented in fig. 3c and fig. 3d and, as expected, show a better comparison with the TROPOMI satellite $SO_2$ VCD retrievals at this particular time. A more detailed vertical emission profile could be constructed using sophisticated inverse modelling techniques (e.g., Eckhardt et al., 2008; Kristiansen et al., 2010; Moxnes et al., 2014), but this is beyond the scope of this paper and is not attempted here.

**Table A1.** The FSS-based timescales estimates for which the NAME simulations have skill compared to TROPOMI at the given spatial resolution N. The given values represent the number of days after start of eruption (rounded by 0.5 days) where the FSS comparison between the NAME simulations and the TROPOMI retrievals drops below 0.5 for various neighbourhood N sizes (value in brackets shows the corresponding model resolution for which the values would be valid), maximum included vertical column density contours and NAME simulations. When calculating the daily FSS value, we only include overpasses where the satellite retrieved total mass was above 0.2 Tg to remove noise. When the model still has skill FSS > 0.6 on the last day where we could determine FSS (17 days), we represent this with a bold number, indicating that the value could be higher than what is presented here.

| N=1 ($0.2° \approx 20$ km) | 0.3 DU | 1 DU | 5 DU | 10 DU | 20 DU | 30 DU | 40 DU | 50 DU |
|---|---|---|---|---|---|---|---|---|
| VolRes1.5 | 17 | 12.5 | 9 | 2.5 | 2 | 1 | 1 | 1 |
| VolRes2.0 | 17 | 12 | 9.5 | 2.5 | 2 | 1.5 | 1 | 1 |
| StratProfile | 17 | 14.5 | 10.5 | 9.5 | 5 | 2 | 1.5 | 1.5 |
| VolRes1.5$_{rd}$ | 17 | 12.5 | 9 | 3 | 2 | 1.5 | 1 | 1 |
| StratProfile$_{rd}$ | 17 | 14.5 | 10.5 | 9.5 | 5 | 2 | 1.5 | 1.5 |
| N=9 ($0.6° \approx 60$ km) | 0.3 DU | 1 DU | 5 DU | 10 DU | 20 DU | 30 DU | 40 DU | 50 DU |
| VolRes1.5 | 17 | 13 | 9 | 4.5 | 2 | 2 | 1.5 | 1.5 |
| VolRes2.0 | **17** | 14 | 10 | 5 | 2.5 | 2 | 2 | 1.5 |
| StratProfile | **17** | 15 | 11 | 9.5 | 5.5 | 4 | 2.5 | 2 |
| VolRes1.5$_{rd}$ | 17 | 13.5 | 9.5 | 4 | 2.5 | 2 | 1.5 | 1.5 |
| StratProfile$_{rd}$ | **17** | 15 | 11.5 | 10 | 5.5 | 4 | 2.5 | 2 |
| N=25 ($1.0° \approx 100$ km) | 0.3 DU | 1 DU | 5 DU | 10 DU | 20 DU | 30 DU | 40 DU | 50 DU |
| VolRes1.5 | **17** | 13.5 | 9 | 5 | 2.5 | 2 | 1.5 | 1.5 |
| VolRes2.0 | **17** | 14 | 10.5 | 5.5 | 2.5 | 2 | 2 | 1.5 |
| StratProfile | **17** | **17** | 11 | 9.5 | 5.5 | 4 | 3 | 2.5 |
| VolRes1.5$_{rd}$ | **17** | 13.5 | 9.5 | 5 | 2.5 | 2 | 1.5 | 1.5 |
| StratProfile$_{rd}$ | **17** | **17** | 11.5 | 10 | 5.5 | 4.5 | 3 | 2.5 |
| N=49 ($1.4° \approx 140$ km) | 0.3 DU | 1 DU | 5 DU | 10 DU | 20 DU | 30 DU | 40 DU | 50 DU |
| VolRes1.5 | **17** | 13.5 | 9 | 5.5 | 2.5 | 2 | 2 | 1.5 |
| VolRes2.0 | **17** | 14.5 | 10.5 | 7 | 3 | 2.5 | 2 | 1.5 |
| StratProfile | **17** | **17** | 11.5 | 10 | 5.5 | 4.5 | 3 | 2.5 |
| VolRes1.5$_{rd}$ | **17** | 14 | 9.5 | 5 | 2.5 | 2 | 2 | 1.5 |
| StratProfile$_{rd}$ | **17** | **17** | 11.5 | 10.5 | 5.5 | 4.5 | 3 | 2.5 |

*Author contributions.* J.dL., A.S. and C.W. designed the project. J.dL. set-up and performed all the NAME simulations and analysed the results. A.S., C.W, M.O. and J.H. contributed to the scientific discussion and interpretation of the results. N.K. provided model input files for the initial NAME simulations. J.dL., N.T. and R.P. were responsible for the preparation of the TROPOMI data for the comparison with the NAME output. I.T. and R.G. processed the IASI data for the comparison. A.S., C.W., and R.G. obtained the funding for this work. J.dL., A.S. and C.W. prepared the manuscript. All the authors reviewed the manuscript.

820

*Competing interests.* The authors do not have any conflict of interests.

*Acknowledgements.* J.dL., A.S., C.W., R.G. and I.T. acknowledge funding from the Natural Environment Research Council (NERC) V-PLUS grants NE/S00436X/1 and NE/S004025/1. I.T. and R.G. are supported by the NERC Centre for Observation and Modelling of Earthquakes, Volcanoes, and Tectonics (COMET). N.T. acknowledges financial support from ESA S5P MPC (4000117151/16/I-LG), Belgium Prodex TRACE-S5P (PEA 4000105598) projects. Funding for PhD work for M.O. was provided by NERC through the University of Exeter, grant NE/M009416/1. Contributions from J.H. and A.S. benefitted from support by the NERC ADVANCE (Aerosol-cloud-climate interactions deduced using Degassing VolcANiC Eruptions), grant NE/T006897/1. The authors would like to thank Helen Webster (Met Office) for helpful discussions on the NAME diffusion parameterisations and Peter Haynes for our discussions on stratospheric mixing. We also like to acknowledge A. Prata and the anonymous reviewer for their comments and suggestions. This work used JASMIN, the UK collaborative data analysis facility (doi:10.1109/BigData.2013.6691556), to run all the simulations and to run the IASI $SO_2$ retrieval. We would like to acknowledge EUMETSAT for providing the IASI spectra; and ECMWF and CEDA for the meteorological profiles used in the IASI retrievals (European Centre for Medium-Range Weather Forecasts, 2012).

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
