# Peer review of "The 2019 Raikoke volcanic eruption: Part 1 Dispersion model simulations and satellite retrievals of volcanic sulfur dioxide"

_Atmospheric Chemistry and Physics, 2020_

## Referee Comment (RC1) · Anonymous Referee #2 · 16 Oct 2020

The 2019 Raikoke volcanic eruption: Part 1 Dispersion model simulations and satellite retrievals of volcanic sulfur dioxide

General comments
In this paper NAME simulations of SO2 are compared to SO2 observations from TROPOMI. It is good to see these new observations being used to evaluate the dispersion of SO2 in a quantitative manner using some interesting spatial verification techniques.  Generally, NAME appears to perform very well, although the peak SO2 concentrations are underestimated. Overall the paper contains some interesting results, but it is long and contains several repetitions and material that is already in the published literature.  The motivation for the experimental design and the hypotheses need to be clearer.

Major comments
1. One aim of the paper is to understand why the observed peak (>20DU) concentrations are underestimated by NAME.  There appears to be 3 potential explanations investigated in the paper. (i) The emission profile – too little SO2 emitted into the stratosphere; (ii) the diffusion parameterisation – too much mixing in the stratosphere; (iii) the tropopause height – too high.  After reading the paper it was not clear which of these explanations was the dominant factor.  I appreciate that it is probably a combination of all 3 but the sensitivity experiments performed seemed a little ad hoc and not best designed to test the 3 hypotheses making it difficult to reach a conclusion. As a result, the abstract contains the results of the sensitivity experiments, but no more general conclusions are/can be included.
2. Given that the focus of the paper is mixing in the stratosphere, it is surprising that there are no references to the extensive literature on this topic.  Below are a few suggestions that should be referred to, but there are many more.

Balluch, M. G., and P. H. Haynes (1997), Quantification of lower stratospheric mixing processes using aircraft data, *J. Geophys. Res.*, **102**, 23,487– 23,504
Colette, A., and G. Ancellet (2006), Variability of the tropospheric mixing and of streamer formation and their impact on the lifetime of observed ozone layers, *Geophys. Res. Lett.*, **33**, L09808
Hall, T. M., and D. Waugh (1997), Tracer transport in the tropical stratosphere due to vertical diffusion and horizontal mixing, *Geophys. Res. Lett.*, **24**, 1383– 1386.
Haynes, P., and J. Anglade (1997), The vertical-scale cascade in atmospheric tracers due to large-scale differential advection, *J. Atmos. Sci.*, **54**, 1121– 1136.
Legras, B., I. Pisso, F. Lefèvre, and G. Berthet (2005), Variability of the Lagrangian turbulent diffusion in the lower stratosphere, *Atmos. Chem. Phys.*, **5**, 1605– 1622.

Minor comments

1. Abstract, line 24.  I'm concerned that the authors are overstating their result.  The paper demonstrates the potential for satellite data to improve the representation of SO2 dispersion for this case study, but I don't think they can claim that it can 'rectify limitations in dispersion models like NAME'.  This would require an analysis of many volcanic SO2 clouds and a systematic improvement.
2. Page 3, line 60. The authors state that 'SO2 clouds are potential tracers for the more difficult observable ash clouds', do they mean 'more difficult to observe ash clouds'?
3. Page 3, line 63. Models are plural so 'is' should be 'are' I think.
4. Page 4, line 80.  What is the 'unprecedented resolution', can the authors be quantitative?
5. Page 5, line 100. Repetition of information from line 79, page 4.

6. Page 5, line 111. Repetition.
7. Page 5, lines 123 and 124. Why are the uncertainties in the SO2 retrievals different?
8. Page 5, line 129. Why is the SO2 layer assumed to be 15km agl?
9. Page 5, line 131. Why are 15km and 7km chosen specifically for sensitivity testing? Can a more extensive systematic sensitivity test be carried out?
10. Page 6, line 136. Please include a reference to justify the 0.3DU threshold used.
11. Page 6, line 140. NAME is a Lagrangian model so doesn't have grid cells. Are you referring to the output grid on which SO2 concentrations are calculated?
12. Page 6, line 143. What is ash-inference? Do the authors mean ash interference?
13. Page 7, line 176. What is the vertical resolution of the meteorological data at the tropopause height?
14. Page 7, lines 177-187. This textbook information is generic to all Lagrangian models. Does it need to be included in the paper?
15. Page 7, line 191. Sigma is the standard deviation of the velocity not the velocity I think.
16. Page 10, table 2. What does full refer to? K_meso?
17. Page 10, table 2. Why is K_meso changed and not K_turb? I couldn't find any motivation for these experiments. Reference to the published literature on stratospheric mixing may help to motivate this experiment.
18. Page 11, line 260. How is the lower stratosphere defined?
19. Page 11, line 262. Why do you need to perform a separate NAME simulation to increase the emissions? Doesn't the output just scale by +33%? Perhaps I have misunderstood this experiment.
20. Page 11, line 264. Repetition of IASI satellite overpasses.
21. Page 11, line 265-270. The motivation for your StratProfile experiment is not clear to me. What hypothesis are you testing? Are you claiming that the first profiles you used were wrong? If so why?
22. Page 11, line 271 and 275. Comparison of the modelled and observed tropopause height suggests that the modelled height may be too high. Therefore, rather than interpreting the emission profile as emitting too little SO2 into the stratosphere, could an alternative interpretation be that the tropopause is too high? Can you perform NAME simulations in which you alter the tropopause height to test this? How does the stability in the stratosphere compare to that in the troposphere? How does the stratospheric stability in NAME compare to that measured by the radiosonde?
23. Page 11, line 283. I couldn't find the part of section 2.1 in which you show that the interference of ash on the retrieval is significantly reduced.
24. Page 11, lines 280-287. There doesn't appear to be any discussion of figure 3c. Does this mean that this figure is not necessary? If so, it should be removed.
25. Page 12, figure 3. It is difficult to see this figure (printed in black and white) due to the coloured background used. Is this necessary?
26. Page 12, figure 3. What does the dashed line represent? Shouldn't it be a box with a latitudinal extent from 48-52N?
27. Page 13, line 295. You conclude that the underestimation is due to an underestimation of the fraction of SO2 released into the stratosphere. Could it also be due to overdispersion of SO2 in the stratosphere?
28. Page 13, 14, 15 and 16. A description of FSS and SAL metrics is already in the published literature. Is it necessary to include this in the main body of the text particularly in this highly idealised form? Since SAL is an object orientated verification metric it would be more informative to show a snapshot of a SO2 cloud with identified objects rather than the idealised example described in the text.
29. Page 13, lines 346-350. For some reason the writing changes to use first person pronouns. I'm not sure what ACP policy is but this section seemed to be written in a different style to the rest of the paper.
30. Page 17, figure 6. Please could the background colour be removed as it's difficult to distinguish the SO2 cloud when printed in black and white.

31. Page 17, line 399. Since the StratProfile has been designed to agree better with the TROPOMI SO2 cloud it's hardly surprising that it does.
32. Page 17, line 404. How do you identify the part of the cloud in the NAME simulation what is within the troposphere and stratosphere using the differences in the simulations? Are you assuming no exchange of SO2 from the stratosphere to the troposphere?
33. Page 21, table 3. This table includes the same information as shown in figure 9 I think but expanded for more thresholds. Could this be moved to the supplementary material?
34. Page 22, figure 10. Why is the SO2 mass lost too fast in the VolRes1.5 simulation? Why is the grey arrow in panel (b) the same as in (a)? Shouldn't it have smaller vertical extent to indicate reduced amplitude spread? Why does the arrow in panel (d) and (d) extend beyond the point with the most negative structure value? Is it necessary to show the individual TROPOMI overpasses? Perhaps including the daily averaged squares only would be easier to follow the evolution in the SAL scores?
35. Page 22, line 471. How do you know that the high VCDs are related to small-scale eddies?
36. Page 23, line 479. The authors state that the S-values are 'relatively close to 0'. Relative compared to what?
37. Page 23, line 480. Does the Location value have units or is it non-dimensional?
38. Page 23, line 483. '4-5 days after the start of the eruption'. Please refer to the dates used in the figure if possible.
39. Page 23, line 490. What happens after 5 days?
40. Page 23, line 493. What is the StratProfile simulation event 'better' than?
41. Page 23, line 499. What is the motivation for reducing the K_meso value by 75%? Is there evidence in the literature that this is appropriate or are you simply using K_meso as a tuning parameter?
42. Page 23, line 504. Why is the S score independent of the diffusion parameter? Is this because the size of the plume becomes greater than the size of the mesoscale eddies that K_meso is representing so the synoptic scale uncertainty dominates?
43. Page 24, figure 11. Here the AAI is used to indicate high concentrations of ash, thereby affecting the TROPOMI SO2 retrievals. This interference is referred to at several points earlier in the paper but until this figure I don't think any evidence was shown to support the statement that ash was potentially contaminating the retrieval. Perhaps this evidence could be included earlier in the paper?
44. Page 24, line 518. Why does the VolRes1.5 simulation loose SO2 mass at a much faster rate than TROPOMI?
45. Page 28, line 641. Do you change K_meso everywhere or just in the stratosphere?
46. Page 28, line 644. Why do you think that a 'precise value for the diffusion parameters' exists? Turbulence is typically patchy, suggesting that a constant value is unsuitable.
47. Page 29, lines 681-684. This appears to be a repetition of the results already stated in the results section, not a conclusion.
48. Page 30, line 695. It is surprising that no reference to the extensive literature on stratospheric dispersion is included here.

---

## Referee Comment (RC2) · Andrew Prata (Referee) · 20 Nov 2020

**General comments**

In this contribution, de Leeuw et al. present satellite observations and dispersion model simulations of the SO2 cloud produced by the 2019 Raikoke (Russia) eruption. The paper mainly uses TROPOMI retrievals to validate NAME SO2 simulations. IASI SO2 height retrievals are also used to assess the SO2 simulations in the vertical and the VolRes team's assessment of the Raikoke eruption is used to constrain the source term (i.e. vertical profile of SO2). The paper is well written and

the figures are excellent. The paper advances our understanding of what the important processes are for large eruptions that inject SO2 at altitudes close to and above the tropopause. Some insights into the NAME SO2 chemistry scheme are given and estimates of the total mass of SO2 injected into the troposphere and stratosphere are provided. Discussion of the e-folding times for the Raikoke event is also presented. This information is particularly important for climate modellers and stratospheric dynamicists as well as VAACs that in the future may be required to provide SO2 forecasts for volcanic eruptions. I recommend publication after addressing some minor revisions suggested below.

**Specific comments**

One theme I noticed throughout the paper was the use of the term 'SO2 concentration' when describing the satellite retrievals. I don't think this is technically correct as the satellite retrievals represent total column densities (VCDs or mass per metre squared). I've highlighted some lines where I think this needs correcting (see Technical corrections below). It is, however, correct to talk about concentrations (mass per metre cubed) when describing the NAME simulations. I think providing units in parentheses would clear up any confusion when discussing these quantities.

In terms of the model setups, one potential issue is the SO2 emission source duration. I see that the authors decided to simulate a constant SO2 emission from 21 June at 18 UTC to 22 June at 3 UTC. However, there's evidence (from Himawari-8) that there were emissions continuing until 22 June at 10 UTC. Therefore, some justification as to why the emission was stopped at 3 UTC is warranted. With regard to the StratProfile simulation and resulting improvements in model skill. This doesn't surprise me too much as the way this profile was derived was based on a fit to the TROPOMI retrievals. Therefore, it should be expected that the simulations would show a better agreement to the TROPOMI retrievals. It would be very interesting to see how the

different model setups (i.e. StratProfile, StratProfile$_{rd}$, VolRes1.5, VolRes2.0) compare to the IASI SO2 retrievals because the TROPOMI and IASI retrievals have different sensitivities. For example, TROPOMI might 'see' SO2 closer to the surface than IASI due to the presence of water vapour or a low thermal contrast. As you compare total column SO2 (VCDs) this difference is not taken into account by the SAL and FSS. I appreciate that comparison against IASI would require a significant amount of extra work and the paper is already rather long; however, adding some discussion on this issue would help the reader appreciate that there are subtleties to be considered when comparing model simulations to satellite retrievals of SO2 from different sensors.

Discussion of the Raikoke SO2 e-folding times. I think it would be worth mentioning e-folding times of similar eruptions to give the Raikoke event some context. For example, Sarychev 2009 erupted at a very similar latitude (also during the NH summer) and there are several papers that discuss e-folding times. Another obvious choice for comparison is Kasatochi 2008.

**Technical corrections/suggestions**

Abstract:
I noticed nothing is said about the SAL-metric in the abstract. It might be worth adding some mention of its use as it's used as a validation metric alongside the FSS.

L11: 'high-concentration regions'. Are the authors referring to vertical column densities here or actual concentrations? This needs to be clarified. For example, Fig. 3 shows VCDs, but Fig. 4 show concentrations.

L11-12: 'NAME shows skill'. Please specify what you mean here by 'skill'. Is it FSS>0.5?

L14: 'high-concentration'. Are you talking about 'concentration' or VCD?

L24: 'high-resolution'. What do you mean by high resolution here? Spatial, temporal, spectral? Please clarify.

L45: 'dormant since 1924'. Add reference. Is this from GVP?

L46: 'erupting until about 0300 UTC'. I'm not sure this is correct. Looking at the Himawari-8 data there are at least two significant explosive eruptions after this time which are contributing SO2 into the plume. Upon close inspection of the Himawari-8 data it looks like eruptive activity is discernible until 1000 UTC. This needs checking as the SO2 simulation results will be affected if the source is stopped at 0300 UTC vs. 1000 UTC.

L49: More examples of satellite observations of the SO2 plume produced by Raikoke have now been published. I provide them here as they may serve as useful references for the authors to consider in their revisions:

Hyman, D. M. and Pavolonis, M. J.: Probabilistic retrieval of volcanic SO2; layer height and partial column density using the Cross-track Infrared Sounder (CrIS), Atmospheric Measurement Techniques, 13(11), 5891–5921, doi:10.5194/amt-13-5891-2020, 2020.

Prata, A. T., Mingari, L., Folch, A., Macedonio, G., and Costa, A.: FALL3D-8.0: a computational model for atmospheric transport and deposition of particles, aerosols and radionuclides – Part 2: model applications, Geosci. Model Dev. Discuss., https://doi.org/10.5194/gmd-2020-166, in review, 2020.

L61: Carn et al. (2009) is another useful reference suggesting the use of SO2

detection as an aviation hazard mitigation tool: Carn, S. A., Krueger, A. J., Krotkov, N. A., Yang, K. and Evans, K.: Tracking volcanic sulfur dioxide clouds for aviation hazard mitigation, Natural Hazards, 51(2), 325–343, doi:10.1007/s11069-008-9228-4, 2009.

L67: Change 'VAACS' to 'VAACs'.

L80: '...measures atmospheric SO2 concentration...'. I suggest changing to '...measures atmospheric SO2 total column densities...'. The TROPOMI product does not provide concentrations (kg m$^{-3}$).

L80: Suggest changing 'unprecedented resolution' to 'unprecedented spatial resolution for UV measurements'.

L97: Change 'sections' to 'section'.

L102: 'SAL-score'. Spell 'SAL' out here as it's the first time in the text this acronym appears.

L129: Change 'TROPOMI VCD data ... is' to 'TROPOMI VCD data ... are'.

L130: 'above ground level'. Is this correct? Aren't the retrievals relative to sea level?

L132: Comparison of the 7 km TROPOMI product doesn't affect the interpretation or overall conclusions. This is somewhat surprising. As, later on, the authors suggest that the height of the SO2 is critical in governing its dispersion. I would have expected a SO2 retrieval assuming a 7 km vs 15 km layer to be significant. This needs some clarification.

L141: Change 'nothern hemisphere' to 'NH' as defined earlier. I would also check all the references to northern hemisphere for consistency if this abbreviation is going to be used.

L142-148: Discussion of factors affecting the SO2 retrievals. What about band saturation issues? Is TROPOMI capable of measuring >1000 DU?

L155: 'different set of assumptions in the retrieval algorithm (e.g. plume height)'. Doesn't TROPOMI also assume a plume height? How is this a different assumption?

Section 2.3: It might be worth emphasising that NAME is a Lagrangian model as later on in the manuscript you talk about releasing 10 million air parcels.

L175: Change 'Global analysis' to 'global analysis'.

L184: Change 'Each NWP' to 'Each NWP model'.

L192: Use of $r$ here to represent a random number. Note that later on $r$ is redefined as the rainfall rate. The symbol might be worth changing to avoid any confusion.

L200: 'SO2 concentrations'. Are we now talking about kg m$^{-3}$ or VCD?

L209: 'After multiplying each grid cell value by the area of the grid cell and summing the resulting mass in each column we obtain the VCDs estimate from NAME'. Please specify the units here. For example, what is the unit of each grid cell value? Is it kg m$^{-3}$? If so, I can't see how multiplying by the area gives you VCD (i.e. kg m$^{-2}$). Wouldn't you need to integrate vertically? I.e. kg m$^{-3}$ to kg m$^{-2}$?

L211: 'which is the detection threshold used for TROPOMI, see section 2.1'. To avoid confusion with the detection threshold of the TROPOMI product (1 DU, that you discuss later) it might be better to say 'which is the detection threshold used for TROPOMI in the present study, see section 2.1'.

L225-229: Check subscripts in chemical reactions.

L239: 'SO2 air concentration'. Please provide units.

L246: 'mass flux'. Is mass flux the correct term here? Usually we talk about the mass eruption rate or mass flow rate (units of kg s$^{-1}$) when referring to ESPs. Mass flux implies units of kg s$^{-1}$ m $^{-2}$. Please check this and in other parts of the manuscript where a mass flux is mentioned.

L252: Mass released between '21 June 18 UTC and 22 June 03 UTC'. Is this correct? As mentioned above, explosive activity was observable beyond 03 UTC and didn't appear to subside until about 10 UTC. How was this end time decided?

L259: 'agl'. I noticed that 'above ground level' is referred to throughout the manuscript. Are you sure this isn't 'above sea level'? i.e. is terrain elevation accounted for in NAME? Also, I thought that the IASI height retrievals are relative to sea level.

L279: 'large amount of ash'. What do you mean by 'large'? Can you provide evidence for this? Perhaps you could refer to the Part 2 paper here.

L316: 'Fractional Skill Score'. Use FSS if you've already defined the acronym. Check this in other places of the manuscript.

L336: 'mass concentrations' or mass loadings? Or VCDs?

L338: 'varying mass densities'. What are the units here?

Figure 5. This is a great illustrative example of how the SAL-score works. Good job.

L369: 'within the domain'. What is the size of the domain used for the SO2 simulations?

L406: 'TROPOMI retrievals'. So all results presented are for the 15 km agl TROPOMI retrievals? It might be worth highlighting this at the beginning of the results section.

L434: Change 'has skill' to 'has skill (FSS$> 0.5$)'.

Figure 10. Which SO2 DU threshold is used here? Please state this in the Figure caption.

L478: 'larger total SO2 mass than the TROPOMI retrievals'. Couldn't this also be due to the spatial coverage of TROPOMI (as you're comparing individual overpasses here)?

L499: Discussion on the differences between StratProfile and StratProfile$_{rd}$ (i.e. Fig. 10c and d). It would be nice to see some S values quoted here or maybe include a table to help understand how large the change in S was when reducing the diffusion parameter by 75%. If you decide to include a table then I suggest reporting values for S, A, L and SAL at key time steps.

Figure 11b. It's difficult to see changes in the SO4 mass deposition. If you want to plot it on the same figure panel then I suggest adding a second y-axis and reducing its range.

L512: Factors affecting TROPOMI estimates of mass. What about band saturation due to high SO2 loads?

L563: 'StratProfile ... compares best with TROPOMI'. What about the StratProfile$_{rd}$? It appears to show better agreement than StratProfile based on Figure 10.

L576: 'able to reasonably accurate simulate'. I'm not sure what this means. Please rephrase.

L597: Discussion on varying input parameters. Did you consider a variation in column height with time? The Raikoke eruption was characterised by a series of explosive eruptions (or 'pulses') that varied in height. Some discussion acknowledging this seems appropriate to add here.

L653: Discussion on Harvey et al. (2018). They were also looking at ash, not SO2, so presumably the impact of the diffusion parameter would be different for that reason as well.

L662: 'high-resolution'. Spatial? Spectral? Temporal? Please clarify.

L698: Change 'In future' to 'In the future'.

---

## Author Comment (AC1) · 12 Apr 2021

**Response to reviews of "The 2019 Raikoke volcanic eruption: Part 1 Dispersion model simulations and satellite retrievals of volcanic sulfur dioxide" by De Leeuw et al.**

We like to thank both reviewers for their time to review the manuscript. We have addressed all reviewer comments in full and revised the manuscript accordingly.

Both reviewers pointed out that the manuscript was long. Therefore, we have attempted to shorten our text where possible by reordering some parts of the manuscript and moving any additional information (e.g. table 3) to the supplementary material to improve the readability of the paper. However, due to the additional discussion that was required to address some specific comments of both reviewers, the final length of the revised manuscript is 600 words longer than the original.

In the remainder of this document, we have written a point-by-point reply to each of the comments from both reviewers. For specific revisions we refer to the revised manuscript.

**Response to review by A. Prata:**

**Specific comments**

*1. One theme I noticed throughout the paper was the use of the term 'SO2 concentration' when describing the satellite retrievals. I don't think this is technically correct as the satellite retrievals represent total column densities (VCDs or mass per metre squared). I've highlighted some lines where I think this needs correcting (see Technical corrections below). It is, however, correct to talk about concentrations (mass per metre cubed) when describing the NAME simulations. I think providing units in parentheses would clear up any confusion when discussing these quantities.*

We thank the reviewer for raising this point and we agree that 'SO$_2$ concentrations' is the wrong term when talking about the satellite retrievals and we have replaced them with VCDs. As suggested by the reviewer, we have also included units to all the sections in the manuscript where it would be beneficial to the reader, also to avoid any confusion.

*2. In terms of the model setups, one potential issue is the SO2 emission source duration. I see that the authors decided to simulate a constant SO2 emission from 21 June at 18 UTC to 22 June at 3 UTC. However, there's evidence (from Himawari-8) that there were emissions continuing until 22 June at 10 UTC. Therefore, some justification as to why the emission was stopped at 3 UTC is warranted.*

We thank the reviewer for pointing out the possible implications of the shorter duration of the eruption in our simulations than what was reported in A. Prata et al. (2020). We based our 21 June 18 UTC - 22 June 3 UTC time window on the VolRes document that was released on the 27th of June. In this document this time window is given as the best estimate for the eruption duration. This was therefore the best information we had available when we performed our simulations, and we feel that this is also the simulation setup that would be used by the community when simulating a similar eruption in near real time. We have updated the text to justify our choice (L.276-281).

[Figure]

*Figure R1: a) SAL score figure for StratProfile simulation but using the longer emission time (21 June 18 UTC - 22 June 10 UTC) and b) the original emission time (subset of points from figure 10c in the manuscript).*

To test the importance of the emission duration, we did some additional short 5-day simulations where we increased the duration of the eruption to the suggested 10 UTC on 22 June, for our VolRes1.5 and StratProfile simulations. Our analysis showed that the overall results are similar (see figure R1), but that the structure of the plume is more diffuse (more mass in the tail of the plume)

and the peak values in the head of the cloud are slightly reduced due to the lower mass eruption rate (same mass emitted over a longer time period). As a result, the comparison with TROPOMI in the SAL-diagram shows that the S-score is higher, but no large differences are found in the A and L score. We only see a possible improvement for 22 June. However due to the large amount of ash in the plume at this point (see figure 11 in manuscript and section 2.1 L148-153), the TROPOMI retrievals are very uncertain and so we should be careful interpreting this single point.

In the new analysis the same large difference remains between the VolRes1.5 and StratProfile simulations that we already observed in our initial simulations. This supports our conclusions that the ratio of $SO_2$ in the upper troposphere/lower stratosphere is more important and that the timing issue raised by the reviewer does not alter our final conclusions of the paper. We therefore decided to keep our initial VolRes duration estimate and added some text to discuss the eruption parameter uncertainties to the discussion section (L.637-645).

***3. With regard to the StratProfile simulation and resulting improvements in model skill. This doesn't surprise me too much as the way this profile was derived was based on a fit to the TROPOMI retrievals. Therefore, it should be expected that the simulations would show a better agreement to the TROPOMI retrievals. It would be very interesting to see how the different model setups (i.e. StratProfile, StratProfilerd, VolRes1.5, VolRes2.0) compare to the IASI SO2 retrievals because the TROPOMI and IASI retrievals have different sensitivities. For example, TROPOMI might 'see' SO2 closer to the surface than IASI due to the presence of water vapour or a low thermal contrast. As you compare total column SO2 (VCDs) this difference is not taken into account by the SAL and FSS. I appreciate that comparison against IASI would require a significant amount of extra work and the paper is already rather long; however, adding some discussion on this issue would help the reader appreciate that there are subtleties to be considered when comparing model simulations to satellite retrievals of SO2 from different sensors.***

The reviewer is correct that the TROPOMI and the IASI retrievals have different sensitivities and that this could influence the comparison between model and satellite data. In our manuscript the IASI data is mainly used as an independent dataset to verify our StratProfile emission profile in terms of height (as this data is not readily available from the TROPOMI satellite) and no attempts are made to compare IASI VCDs with NAME in detail due to many additional assumptions and (error) analysis that would have to be considered and would be a full study on its own (see L.159-168). We agree that it would be an interesting exercise to do more analysis on the IASI comparison, but this is outside the scope of this current work.

Finally, we would also like to point out that the StratProfile simulation is only using a prescribed emission profile that is based on TROPOMI for 23 June. Therefore, getting the best comparison on longer timescales for these simulations is not a trivial result (see also point 31 of reviewer #2).

***4. Discussion of the Raikoke SO2 e-folding times. I think it would be worth mentioning e-folding times of similar eruptions to give the Raikoke event some context. For example, Sarychev 2009 erupted at a very similar latitude (also during the NH summer) and there are several papers that discuss e-folding times. Another obvious choice for comparison is Kasatochi 2008.***

We thank the reviewer for this suggestion and have included e-folding time estimates for the Sarychev 2009 and Kasatochi 2008 eruptions to the manuscript to put the Raikoke estimates into context (L.558-561).

**Technical corrections/suggestions**

**Abstract:**
**1. I noticed nothing is said about the SAL-metric in the abstract. It might be worth adding some mention of its use as it's used as a validation metric alongside the FSS.**

We thank the reviewer for the suggestion and have now introduced the SAL-metric in the abstract.

**2. L11: 'high-concentration regions'. Are the authors referring to vertical column densities here or actual concentrations? This needs to be clarified. For example, Fig.3 shows VCDs, but Fig. 4 shows concentrations.**

We are discussing vertical column densities here and we have changed the text accordingly.

**3. L11-12: 'NAME shows skill'. Please specify what you mean here by 'skill'. Is it FSS>0.5?**

When discussing the skill of the NAME model, this is a combination of the SAL-score and the FSS score. Both metrics and their interpretation are explained in detail in the manuscript (sections 2.4, 3.2 and 3.3). We disagree with the reviewer that this should be specified in the abstract, as this information is too detailed and would distract from the main message.

**4. L14: 'high-concentration'. Are you talking about 'concentration' or VCD?**

We are discussing VCDs here and we have changed the text accordingly.

**5. L24: 'high-resolution'. What do you mean by high resolution here? Spatial, temporal, spectral? Please clarify.**

High spatial resolution, which we have now clarified in the document

**6. L45: 'dormant since 1924'. Add reference. Is this from GVP?**

Yes, this information was taken from a GVP report (Crafford and Venzke, 2019). We have now included two references in the manuscript to this report that state this information (L.46-47)

**7. L46: 'erupting until about 0300 UTC'. I'm not sure this is correct. Looking at the Himawari-8 data there are at least two significant explosive eruptions after this time which are contributing SO2 into the plume. Upon close inspection of the Himawari-8 data it looks like eruptive activity is discernible until 1000 UTC. This needs checking as the SO2 simulation results will be affected if the source is stopped at 0300 UTC vs. 1000 UTC.**

We have based this on the initial analysis by the Volcanic Response team. We refer the reviewer to the second specific comment (page 2-3) of this review for more details.

**8. L49: More examples of satellite observations of the SO2 plume produced by Raikoke have now been published. I provide them here as they may serve as useful references for the authors to consider in their revisions:**
**Hyman, D. M. and Pavolonis, M. J.: Probabilistic retrieval of volcanic SO2; layer height and partial column density using the Cross-track Infrared Sounder (CrIS), Atmospheric Measurement Techniques, 13(11), 5891–5921, doi:10.5194/amt-13-**

**5891-2020, 2020.**
**Prata, A. T., Mingari, L., Folch, A., Macedonio, G., and Costa, A.: FALL3D-8.0:**
**a computational model for atmospheric transport and deposition of particles, aerosols**
**and radionuclides – Part 2: model applications, Geosci. Model Dev. Discuss.,**
**https://doi.org/10.5194/gmd-2020-166, in review, 2020.**

We thank the reviewer for these references, and we have now included them to our manuscript.

**9. L61: Carn et al. (2009) is another useful reference suggesting the use of SO2
detection as an aviation hazard mitigation tool: Carn, S. A., Krueger, A. J., Krotkov, N.
A., Yang, K. and Evans, K.: Tracking volcanic sulfur dioxide clouds for aviation hazard
mitigation, Natural Hazards, 51(2), 325–343, doi:10.1007/s11069-008-9228-4, 2009.**

We thank the reviewer for this suggestion and have included this reference.

**10. L67: Change 'VAACS' to 'VAACs'.**

Thank you for spotting this.

**11. L80: '...measures atmospheric SO2 concentration...'. I suggest changing to '...measures
atmospheric SO2 total column densities...'. The TROPOMI product does not
provide concentrations (kg m–3).**

Agreed and changed in the manuscript.

**12. L80: Suggest changing 'unprecedented resolution' to 'unprecedented spatial
resolution for UV measurements'.**

Done as suggested by the reviewer.

**13. L97: Change 'sections' to 'section'.**

Done

**14. L102: 'SAL-score'. Spell 'SAL' out here as it's the first time in the text this acronym appears.**

Done as suggested by the reviewer.

**15. L129: Change 'TROPOMI VCD data ... is' to 'TROPOMI VCD data ... are'.**

Done

**16. L130: 'above ground level'. Is this correct? Aren't the retrievals relative to sea
level?**

Yes, the reviewer is correct, and we have changed the references to the TROPOMI retrieval heights to above sea level.

**17. L132: Comparison of the 7 km TROPOMI product doesn't affect the interpretation or overall
conclusions. This is somewhat surprising. As, later on, the authors suggest that the height of the**

**SO2 is critical in governing its dispersion. I would have expected a SO2 retrieval assuming a 7 km vs 15 km layer to be significant. This needs some clarification.**

We agree with the reviewer that the SO$_2$ retrievals assuming a 7 km or a 15 km layer will result in a different absolute value for the VCDs one obtains from TROPOMI and (through the application of the AK) also a different value of the VCD from the NAME simulations. As a result, the values presented in for example figure 11 would be different. However, we found that both the NAME results and the TROPOMI retrievals both shifted in the same direction (in the 7 km case, the application of AKs to NAME accounts for the different sensitivity to layers higher in the stratosphere), leaving very similar results in terms of the relative difference between the two datasets and leading to very similar SAL-scores and FSS scores (see for example figure R2).

[Figure]

*Figure R2: a) The SAL score figure for StratProfile simulation using the 7 km AK TROPOMI product. b) The same as panel a), but using the 15 km AK product.*

The emission profile used in NAME does have a large impact on how the SO$_2$ cloud disperses in the atmosphere and is not influenced by the TROPOMI SO$_2$ retrieval. Both the 7 km vs the 15 km layer retrievals shows that we get better results when we emit more mass into the stratosphere.

**18. L141: Change 'northern hemisphere' to 'NH' as defined earlier. I would also check all the references to northern hemisphere for consistency if this abbreviation is going to be used.**

Thank you for the suggestion and we have changed all the northern hemisphere references to NH.

**19. L142-148: Discussion of factors affecting the SO2 retrievals. What about band saturation issues? Is TROPOMI capable of measuring >1000 DU?**

For very high SO$_2$ vertical column densities, the TROPOMI algorithm switches to a wavelength range (window 3: 360-390nm) that contains weak SO$_2$ absorption bands. For SO$_2$ VCDs ~1000 DU, the measured optical depth would be about 0.05 (according to Fig 3 of Theys et al. 2017), which is well below any possible non-linear spectral response. Therefore, the main factor affecting the TROPOMI SO$_2$ VCDs is not band saturation due to SO$_2$. High SO$_2$ column values often coincide with high ash loadings. In particular, for strongly absorbing aerosols (like volcanic ash) emitted in the same vertical layers as SO$_2$, the light penetration in the plume is strongly affected by aerosols and the sensitivity of the satellite retrievals is limited to the top of the plume. During the first 1 day of the Raikoke eruption, the TROPOMI SO$_2$ retrieval will strongly underestimate the true VCD (it can be up to a factor of 5) because of the high ash loadings. This aspect is discussed in the manuscript (L.147-153).

**20. L155: 'different set of assumptions in the retrieval algorithm (e.g. plume height)'. Doesn't TROPOMI also assume a plume height? How is this a different assumption?**

We refer here to the assumption of the actual height used by the two satellite retrievals, not the concept of plume height. We have some data available for the IASI data, but part of the IASI data is the height retrieval and is therefore not assumed to be at 7 or 15 km like in our TROPOMI product. Therefore, the comparison would not be straightforward and is omitted here. We have updated the manuscript to make this clearer (L.160-166).

**21. Section 2.3: It might be worth emphasising that NAME is a Lagrangian model as later on in the manuscript you talk about releasing 10 million air parcels.**

Thank you for the suggestion and we have included it into the manuscript (L. 170)

**22. L175: Change 'Global analysis' to 'global analysis'.**

Done

**23. L184: Change 'Each NWP' to 'Each NWP model'.**

Done

**24. L192: Use of r here to represent a random number. Note that later on r is redefined as the rainfall rate. The symbol might be worth changing to avoid any confusion.**

Agreed and we have now changed it to the symbol d

**25. L200: 'SO2 concentrations'. Are we now talking about kg m−3 or VCD?**

We are talking about kg $m^{-3}$ and so $SO_2$ concentrations is the correct terminology here. We have added the units to the manuscript for clarification (L.223)

**26. L209: 'After multiplying each grid cell value by the area of the grid cell and summing the resulting mass in each column we obtain the VCDs estimate from NAME'. Please specify the units here. For example, what is the unit of each grid cell value? Is it kg m−3? If so, I can't see how multiplying by the area gives you VCD (i.e. kg m−2). Wouldn't you need to integrate vertically? I.e. kg m−3 to kg m−2?**

We agree with the reviewer that the current description is not detailed enough. We indeed calculate the VCD values by vertically integrating the $SO_2$ mass concentrations (kg $m^{-3}$). We have changed the text by adding units and made the statement more precise. (L.231)

**27. L211: 'which is the detection threshold used for TROPOMI, see section 2.1'. To avoid confusion with the detection threshold of the TROPOMI product (1 DU, that you discuss later) it might be better to say 'which is the detection threshold used for TROPOMI in the present study, see section 2.1'.**

We agree with the reviewer that some clarification is needed and have updated the text (L.234).

**28. L225-229: Check subscripts in chemical reactions.**

Thank you for this comment and we have updated the subscripts in the chemical reactions.

**29. L239: 'SO2 air concentration'. Please provide units.**

This should be the unit ($kg\ m^{-3}$) and is added to the manuscript. (L.263)

**30. L246: 'mass flux'. Is mass flux the correct term here? Usually we talk about the mass eruption rate or mass flow rate (units of $kg\ s^{-1}$) when referring to ESPs. Mass flux implies units of $kg\ s^{-1}\ m^{-2}$. Please check this and in other parts of the manuscript where a mass flux is mentioned.**

We agree that the term 'mass flux' is poorly chosen and have changed it throughout the manuscript to mass eruption rate.

**31. L252: Mass released between '21 June 18 UTC and 22 June 03 UTC'. Is this correct? As mentioned above, explosive activity was observable beyond 03 UTC and didn't appear to subside until about 10 UTC. How was this end time decided?**

We have based this on the initial analysis by the Volcanic Response team. We refer the reviewer to the second specific comment (page 2-3) for more details.

**32. L259: 'agl'. I noticed that 'above ground level' is referred to throughout the manuscript. Are you sure this isn't 'above sea level'? i.e. is terrain elevation accounted for in NAME? Also, I thought that the IASI height retrievals are relative to sea level.**

The output from IASI is indeed relative to sea level, while our output for the NAME simulations was above ground level. For consistency, we have now converted all the NAME results to heights above sea level to avoid confusion and removed all the references to above ground level throughout the entire manuscript.

**33. L279: 'large amount of ash'. What do you mean by 'large'? Can you provide evidence for this? Perhaps you could refer to the Part 2 paper here.**

We have changed the paragraph which removed this sentence. Instead, we have included the ash estimate from the part 2 paper (15 Tg) in a different paragraph of the same section (L.306).

**34. L316: 'Fractional Skill Score'. Use FSS if you've already defined the acronym. Check this in other places of the manuscript.**

Agreed with the reviewer and we have replaced most of the references to the acronyms. We have decided to spell out the full term once more in the conclusions section to clarify the acronym again to help the reader.

**35. L336: 'mass concentrations' or mass loadings? Or VCDs?**

We used VCDs here and have updated the text to represent the correct units and variables (L.370-371).

**36. L338: 'varying mass densities'. What are the units here?**

See previous point, we have now changed it to the VCD, as this is the relevant unit for our study.

**37. Figure 5. This is a great illustrative example of how the SAL-score works. Good job.**

Thank you for your kind remark.

**38. L369: 'within the domain'. What is the size of the domain used for the SO2 simulations?**

The domain for the simulations is the Northern Hemisphere > 25° N. For clarity, we have included this in the manuscript when discussing the setup of our simulations in the methodology section (L.274). We have not included it in the section suggested here, as this is discussing the schematic of figure 5 and we feel that this would confuse the reader.

**39. L406: 'TROPOMI retrievals'. So all results presented are for the 15 km agl TROPOMI retrievals? It might be worth highlighting this at the beginning of the results section.**

Thank you for the suggestion. We think that the additional information presented in the methods section (L.128-136) clearly describes what results are presented in this manuscript. To avoid repetition, we have decided not to repeat this information at the start of the results section.

**40. L434: Change 'has skill' to 'has skill (FSS> 0.5)'.**

Done (L.468)

**41. Figure 10. Which SO2 DU threshold is used here? Please state this in the Figure caption.**

We have used 0.3 DU as a threshold for TROPOMI data and no threshold for the NAME data, as described in section 2.3.2. We have updated the caption of the figure to clarify this for the reader.

**42. L478: 'larger total SO2 mass than the TROPOMI retrievals'. Couldn't this also be due to the spatial coverage of TROPOMI (as you're comparing individual overpasses here)?**

We thank the reviewer for his comment, but we think our statement is correct. When comparing individual TROPOMI overpasses, we also only select the NAME output that corresponds with the spatial coverage of the TROPOMI satellite. Therefore, the comparison is over the same part of the plume for each of the overpasses and any positive S value is due to the presence of more $SO_2$ in the NAME simulation in the single TROPOMI overpass domain. We have updated the text (L.390-392) to clarify that we compare only the NAME output across each TROPOMI overpass when calculating the individual overpass SAL scores.

**43. L499: Discussion on the differences between StratProfile and StratProfilerd (i.e. Fig. 10c and d). It would be nice to see some S values quoted here or maybe include a table to help understand how large the change in S was when reducing the diffusion parameter by 75%. If you decide to include a table then I suggest reporting values for S, A, L and SAL at key time steps.**

We thank the reviewer for the suggestion and have added some values for S in the text (L.536). All the SAL-value data can be found in the data files stored in the data repository (link to the files is provided in the paper). In the paper we are not aiming to provide a 'best value' for the reduction of the diffusion parameter in general, as the 'optimal' parameter will very likely be different for each eruption. The main result is that a reduction of the diffusion parameter gives better S values, but it is less important to know the precise values in terms of improvement for each timestep. With this in mind and the data being available in the repository, we have decided not to include an additional table.

**44. Figure 11b. It's difficult to see changes in the SO4 mass deposition. If you want to plot it on the same figure panel then I suggest adding a second y-axis and reducing its range.**

We have tried to add an additional y-axis to the panel but found that the resulting figure was harder to describe and as a result harder to interpret. For the discussion presented in the paper, it is important to see that these values are small (L.577) rather than knowing the precise evolution of the mass deposition. If the reviewer is interested in the precise values, these can be found in the provided data as described in the data availability section of the paper.

**45. L512: Factors affecting TROPOMI estimates of mass. What about band saturation due to high SO2 loads?**

This comment is covered by our response to comment 19.

**46. L563: 'StratProfile ... compares best with TROPOMI'. What about the StratProfilerd? It appears to show better agreement than StratProfile based on Figure 10.**

We thank the reviewer for this comment. Both simulations will have the same result in terms of the mass burden evolution, as we emit the same mass at the same altitudes. We have now updated the text to clarify this point more clearly (L.606).

**47. L576: 'able to reasonably accurate simulate'. I'm not sure what this means. Please rephrase.**

We agree with the reviewer that the current wording is poor. We have changed the wording to (L.619):
*'is able to simulate the 3D structure of the volcanic SO2 cloud (and consequently the SO4 cloud) for the 2019 Raikoke eruption.'*

**48. L597: Discussion on varying input parameters. Did you consider a variation in column height with time? The Raikoke eruption was characterised by a series of explosive eruptions (or 'pulses') that varied in height. Some discussion acknowledging this seems appropriate to add here.**

We agree with the reviewer and have added some discussion to the manuscript (L.637-645)

**49. L653: Discussion on Harvey et al. (2018). They were also looking at ash, not SO2, so presumably the impact of the diffusion parameter would be different for that reason as well.**
This is a good point and we have added this to our discussion section (L.716).

**50. L662: 'high-resolution'. Spatial? Spectral? Temporal? Please clarify.**

We refer here to spatial resolution and have updated the text to clarify (L.743).

**51. L698: Change 'In future' to 'In the future'.**

Done

**Response to Referee #2:**

**Major comments:**

*1. One aim of the paper is to understand why the observed peak (>20DU) concentrations are underestimated by NAME. There appears to be 3 potential explanations investigated in the paper. (i) The emission profile – too little SO2 emitted into the stratosphere; (ii) the diffusion parameterisation – too much mixing in the stratosphere; (iii) the tropopause height – too high. After reading the paper it was not clear which of these explanations was the dominant factor. I appreciate that it is probably a combination of all 3 but the sensitivity experiments performed seemed a little ad hoc and not best designed to test the 3 hypotheses making it difficult to reach a conclusion. As a result, the abstract contains the results of the sensitivity experiments, but no more general conclusions are/can be included.*

The main aim of the paper was to determine the skill of NAME to represent the dispersion of volcanic $SO_2$ in the atmosphere. When we compared the VolRes1.5 NAME output with the TROPOMI retrievals, we found that the overall comparison is good, but as the reviewer pointed out, the dispersion model is not capturing the observed high vertical column densities (VCD) and had a different extent of the $SO_2$ cloud (e.g., large positive S-score) when compared to TROPOMI. To understand what factors could contribute to these discrepancies, we have investigated two main potential factors: (i) emission profile and (ii) changing the mixing in the troposphere and stratosphere. Point (iii) is not investigated in the paper, see also answer to comment 22. By changing (i), we found that we can get a much better comparison when we emit more mass into the stratosphere (possibly linked to radiative lofting, see comment 21 on page 15 of this document). Furthermore, we see an improvement in capturing the horizontal extent of the cloud when we reduced the horizontal mixing (K_meso) parameterisation in the model for the free atmosphere (point ii), hinting that the current parameterisation is not optimal for Upper Troposphere /Lower Stratosphere (UT/LS) mixing in the atmosphere.

We are, however, not aiming to simulate the observed small-scale peak VCD features, as we know that the NAME simulations are limited by the resolution of the wind field input. Smaller scale atmospheric motions (e.g., meso-scale eddies) are represented by random small perturbations in NAME (see section 2.3.1). Therefore, our model simulations will only represent one possible realisation of the actual motions occurring in the atmosphere and due to the randomness of the motion will ultimately reduce peak VCD values in the simulations. As a result, we will always have more diffusion in the model, and we won't be able to capture the small-scale peak values as retrieved by TROPOMI (see also L.445-449).

As a result, we have not added any more details on the underestimation of the very high peak values in the paper as it would distract the reader from the main conclusions. We have updated the text in abstract to better reflect the nature of our main findings.

*2. Given that the focus of the paper is mixing in the stratosphere, it is surprising that there are no references to the extensive literature on this topic. Below are a few suggestions that should be referred to, but there are many more. (We haven't added these to the response document, but they can be found in the original reviewers' comments)*

We thank the reviewer for their suggested literature on stratospheric mixing. We agree that part of this literature should be included in the manuscript. We have now added a short discussion of the relevant literature on stratospheric mixing to section 2.3.1 (L.206-221), the discussion section (L.701) and our conclusions section (L.758).

**Minor comments:**

*1. Abstract, line 24. I'm concerned that the authors are overstating their result. The paper demonstrates the potential for satellite data to improve the representation of SO2 dispersion for this case study, but I don't think they can claim that it can 'rectify limitations in dispersion models like NAME'. This would require an analysis of many volcanic SO2 clouds and a systematic improvement.*

We agree with the reviewer that we cannot claim that our work is enough to rectify limitations of dispersion models in general. In order to do this, one would indeed need to look at a large set of eruptions and multiple models. Our work for example shows that the model is in a better agreement with the satellite observations when we reduce the horizontal diffusion parameters at the height of the volcanic cloud within the model. While this is only one case study, it shows the potential of combining high resolution satellite observations and dispersion model output to find potential shortcomings of the models and rectify these limitations. To reflect this better, we have changed the wording to:

*'Our work illustrates how the synergy between high-resolution satellite retrievals and dispersion models can identify potential limitations of dispersion models like NAME, which will ultimately help to improve dispersion modelling efforts of volcanic SO$_2$ clouds.'*

*2. Page 3, line 60. The authors state that 'SO2 clouds are potential tracers for the more difficult observable ash clouds', do they mean 'more difficult to observe ash clouds'?*

Yes, the previous wording was not correct, and we have changed the wording (L.64).

*3. Page 3, line 63. Models are plural so 'is' should be 'are' I think.*

Agreed

*4. Page 4, line 80. What is the 'unprecedented resolution', can the authors be quantitative?*

The values are stated in the method section on the TROPOMI data (L.120) and are shown in comparison with previous satellite products in figure 2 from Theys et al 2019. For completeness, we have now also included the TROPOMI resolution in the introduction (L.84).

*5. Page 5, line 100. Repetition of information from line 79, page 4.*

We are unclear on the repetition that the reviewer is referring to and would like to ask for clarification.

*6. Page 5, line 111. Repetition.*

We have removed the mention of the ESA's S5P satellite from the introduction to avoid the repetition.

*7. Page 5, lines 123 and 124. Why are the uncertainties in the SO2 retrievals different?*

The uncertainty of +/-50% is a typical value applicable to various vertical distributions of SO$_2$, from boundary layer (including anthropogenic pollution cases) to stratospheric injection. For the latter, the uncertainty is arguably lower and in the order of +/-30%. Note that these uncertainties are for

standard conditions with reasonably low AODs. For freshly emitted ash-laden plumes, the uncertainty on $SO_2$ VCD can be much larger (read also comment 19 of the review by A. Prata).

**8. Page 5, line 129. Why is the SO2 layer assumed to be 15km agl?**

Our choice is due to the available data provided when downloading the TROPOMI satellite product. The sensitivity of the TROPOMI instrument to the height of the $SO_2$ cloud is the main uncertainty in the satellite product (see for example Theys, 2018). However, this dependency works in tandem with all the other assumptions (e.g. cloud fraction, relative humidity, ozone concentration) when calculating the satellite VCD estimate. Therefore, for each additional height-scenario that is considered, this recalculation of all the variables (e.g. Averaging Kernel) would be time-consuming. As a compromise, the TROPOMI product is made available for three $SO_2$ layer height scenario's: 1, 7 and 15 km asl, representing an effusive, medium or an explosive volcanic eruption respectively.

As a result, the Averaging Kernel values based on the TROPOMI data are also only calculated for these three scenarios (1, 7 and 15 km) and so we are limited in our study to one of these three choices. Based on both the VolRes and the StratProfile profile, we assumed the best comparison to be for 15 km. But as stated the 7 km data, although giving different absolute VCD values, lead to the same conclusions as presented in the paper. (see also our figure provided to comment 17 from review by A.Prata)

We have changed the text to better explain our assumption (L128-L136).

**9. Page 5, line 131. Why are 15km and 7km chosen specifically for sensitivity testing? Can a more extensive systematic sensitivity test be carried out?**

See previous comment.

**10. Page 6, line 136. Please include a reference to justify the 0.3DU threshold used.**

We have included a reference to the 'S5P Mission Performance Centre Sulphur Dioxide readme' document (Theys et al. (2020)), where the threshold of 0.3 DU is presented in table 1 as the random error for the lower stratosphere-upper troposphere $SO_2$ product.

**11. Page 6, line 140. NAME is a Lagrangian model so doesn't have grid cells. Are you referring to the output grid on which SO2 concentrations are calculated?**

Yes. We have updated the text to clearly state 'output grid' (L.144).

**12. Page 6, line 143. What is ash-inference? Do the authors mean ash interference?**

Yes, this is a typo. We indeed mean ash interference and have updated the text (L.148).

**13. Page 7, line 176. What is the vertical resolution of the meteorological data at the tropopause height?**

The vertical resolution is approximately 22 hPa at the tropopause height (the vertical resolution of the meteorological data varies with height), which corresponds to approximately 600 m. We have included this value in the manuscript. (L.186)

**14. Page 7, lines 177-187. This textbook information is generic to all Lagrangian**

*models. Does it need to be included in the paper?*

An important part of the manuscript investigates the importance of the diffusion scheme and its impact on the perturbation factor (u'). Some of this work might be highly relevant to other areas of the scientific community (e.g., the satellite retrieval) that are not familiar with Lagrangian models in general. Therefore, we think this short summary is an essential part of the manuscript to describe the main features of a Lagrangian model to help the non-expert readers to interpret our results.

*15. Page 7, line 191. Sigma is the standard deviation of the velocity not the velocity I think.*

Yes, you are correct that sigma represents the standard deviation of the velocity. We have updated the text accordingly (L.201-202).

*16. Page 10, table 2. What does full refer to? K_meso?*

The term 'full' represents the K_meso values used in the 'standard' NAME setup (presented in table 1). To clarify this in the paper, we have removed the term 'full' and wrote down K_meso instead and updated the table 2 caption.

*17. Page 10, table 2. Why is K_meso changed and not K_turb? I couldn't find any motivation for these experiments. Reference to the published literature on stratospheric mixing may help to motivate this experiment.*

The main reason for not changing the K_turb factor is that we found in some initial tests that it has only a very minor influence on the results (as also noted by the small values of this parameter in table 1). Furthermore, we are only able to investigate the horizontal diffusion, as the satellite does not contain information on the vertical structure. Therefore, we can't investigate the accuracy of the vertical turbulent mixing using the TROPOMI product and we therefore have no justification to change the vertical mixing parameters.

For more details/arguments on the K_meso values presented in our paper, please see our reply to comment 41. We have added more information on our choices for changing K_meso and not K_turb in section 2.3.1 (L.218-221).

*18. Page 11, line 260. How is the lower stratosphere defined?*

Anything above the tropopause (11 km) and 18 km asl. We have updated the text to make this point clear to the reader (L.298).

*19. Page 11, line 262. Why do you need to perform a separate NAME simulation to increase the emissions? Doesn't the output just scale by +33%? Perhaps I have misunderstood this experiment.*

The experiment is needed as also the chemical conversion and the rate of wet and dry deposition are influenced by the concentrations of $SO_2$. Therefore, this process is not linear. We have updated the text (L.330-332) to clarify the need for this additional experiment.

*20. Page 11, line 264. Repetition of IASI satellite overpasses.*

Removed from the manuscript to avoid repetition.

*21. Page 11, line 265-270. The motivation for your StratProfile experiment is not clear to me. What hypothesis are you testing? Are you claiming that the first profiles you used were wrong? If so why?*

We have addressed this comment in section 2.3.4 (L.275-328) and in the discussion section 4 (L.630-656). The reason for deriving additional emission profiles (aside from the one obtained from the VolRes team) is to investigate potential emission source parameter uncertainties.

The Raikoke eruption resulted in a mixed-phase (ash, chemical aerosol and gas) plume near the tropopause. In the study by Muser et al. (2020) it is suggested that the radiative lofting effect for this eruption is of the order of 2-3 km after 36 hours for the ash plume. Although the details of the mixing of the ash and sulfate and $SO_2$ are not known, it is not unreasonable to assume similar lofting values for the $SO_2$ cloud. The NAME model does not account for the radiative lofting effect that took place due to the presence of mixtures of ash and $SO_4$ in the initial plume. Therefore, using the VolRes1.5 emission profile (which was derived during the first hours after the eruption) could lead to $SO_2$ at the wrong altitudes and therefore it is more appropriate to initialise NAME with an 'effective' emission profile 1-2 days after the initial emission time.

We want to stress here that we do not suggest the VolRes profile is wrong, but it will lead to a different and probably less accurate $SO_2$ cloud dispersion simulation when the dispersion model (like NAME) doesn't account for radiative lofting due to absorbing species in the volcanic cloud. Therefore, we derive an additional profile, called StratProfile, from data 30 hours after the start of the eruption when ash interference is smaller, and also most radiative lofting has taken place.

*22. Page 11, line 271 and 275. Comparison of the modelled and observed tropopause height suggests that the modelled height may be too high. Therefore, rather than interpreting the emission profile as emitting too little SO2 into the stratosphere, could an alternative interpretation be that the tropopause is too high? Can you perform NAME simulations in which you alter the tropopause height to test this? How does the stability in the stratosphere compare to that in the troposphere? How does the stratospheric stability in NAME compare to that measured by the radiosonde?*

We thank the reviewer for this comment, but unfortunately this would be very difficult to test as the NAME model is offline and is forced by the wind field obtained from the MetUM model. Therefore, all our results (including the tropopause height) simulated by NAME are dictated by the stability in the MetUM NWP output and cannot be altered within the NAME model itself.

As a result, NAME is not suitable to investigate the atmospheric stability and the potential errors related to the misrepresentation of the tropopause height in the model. One would have to rerun the MetUM NWP simulations with different settings that could change the tropopause height and the resulting wind fields. However, this would require an enormous amount of work that is not feasible for this manuscript and therefore is not attempted here.

*23. Page 11, line 283. I couldn't find the part of section 2.1 in which you show that the interference of ash on the retrieval is significantly reduced.*

We refer here to the AAI index, which is much lower for the second day as mentioned in section 2.1. We have now changed the text to better represent this point and we only go into the details for the Raikoke eruption in section 3.4 to avoid confusion.

**24. Page 11, lines 280-287. There doesn't appear to be any discussion of figure 3c. Does this mean that this figure is not necessary? If so, it should be removed.**

We have added figure 3c to show the new VCDs, which show a much better comparison between the satellite retrieval and the NAME simulation using the StratProfile, which is discussed in section 3.1.

**25. Page 12, figure 3. It is difficult to see this figure (printed in black and white) due to the coloured background used. Is this necessary?**

We have now replaced figure 3 with a new version where we use a grey and white background.

**26. Page 12, figure 3. What does the dashed line represent? Shouldn't it be a box with a latitudinal extent from 48-52N?**

Yes, the reviewer is correct, and we have added a box to the new version of figure 3.

**27. Page 13, line 295. You conclude that the underestimation is due to an underestimation of the fraction of SO2 released into the stratosphere. Could it also be due to overdispersion of SO2 in the stratosphere?**

We disagree with this statement, as we can see that the horizontal structure of the cloud in figures 3a and 3b is not too different. Therefore, the small differences that might be related to the change in horizontal diffusion won't be able to account for the strong decrease in $SO_2$ VCDs in this region of the cloud. When we investigated the simulations with the reduced horizontal diffusion, we found that the VCDs in the VolRes1.5 simulation were still too small in this part of the cloud, excluding that overdispersion is the main source for the underestimate. Note also that in our manuscript we only investigate the impact of horizontal diffusion (see comments 17 and 41). We have now updated the manuscript to include a short discussion of this point (L325-327).

**28. Page 13, 14, 15 and 16. A description of FSS and SAL metrics is already in the published literature. Is it necessary to include this in the main body of the text particularly in this highly idealised form? Since SAL is an object orientated verification metric it would be more informative to show a snapshot of a SO2 cloud with identified objects rather than the idealised example described in the text.**

We thank the reviewer for the suggestion, but we have decided to keep the current description in the main body of the text. The review by A. Prata was positive about this section (see comment 37 of his review) and we found that by removing the description we would make it harder for the reader to interpret the SAL-diagrams we present later in the manuscript.

**29. Page 13, lines 346-350. For some reason the writing changes to use first person pronouns. I'm not sure what ACP policy is but this section seemed to be written in a different style to the rest of the paper.**

We would like to ask the reviewer for a clarification of this comment, as we cannot identify the inconsistency in writing style mentioned by the reviewer.

**30. Page 17, figure 6. Please could the background colour be removed as it's difficult to distinguish the SO2 cloud when printed in black and white.**

We have updated the figure to remove the coloured background. For consistency, we also removed the background colour from figure 7.

**31. Page 17, line 399. Since the StratProfile has been designed to agree better with the TROPOMI SO2 cloud it's hardly surprising that it does.**

We thank the reviewer for this comment, but we disagree with their assessment of the StratProfile being designed to better agree with TROPOMI for the full simulation presented in the paper. For our StratProfile simulations, we only obtain the initial eruption emission profile from the TROPOMI retrievals on 23 June and run the NAME model independently of the satellite retrievals afterwards.

We agree that it is no surprise that it performs well during the start of the simulation, as it is designed to best fit TROPOMI during the first day of the eruption (as is shown and written in this section of the paper). However, it is not a trivial result that the full simulation would also be the best fit to the TROPOMI satellite retrievals, as many other factors (e.g. simulated wind field, radiative heating, mixing) can influence the dispersion of the SO$_2$ cloud, which could lead to a much weaker comparison if they are wrongly represented in NAME. Therefore, it would be possible that after a week, better results would be obtained when using the VolRes1.5 or VolRes2.0 profile.

Finally, using observations, including satellite retrievals, to optimise the eruption parameters is also how the VAACs would operate in a near-real time situation to obtain the best possible setup for their NAME simulations. In our example we show that having correct source parameters is essential for multi-phase plumes near the tropopause, as small changes will determine the accuracy of your simulation for both short and long simulations. We have updated the text to better represent this argument in the document when discussing the mass burden evolution (L.568-573).

**32. Page 17, line 404. How do you identify the part of the cloud in the NAME simulation that is within the troposphere and stratosphere using the differences in the simulations? Are you assuming no exchange of SO2 from the stratosphere to the troposphere?**

Yes, it is true that this is not a qualitative measure. The difference between the two emission profiles is mainly the amount emitted into the stratosphere/troposphere. We indeed assume that the exchange between the stratosphere and troposphere is limited and hence the differences we see are related to the part emitted in the stratosphere/troposphere. We are not making claims about the absolute values, but merely point out that the colour patterns are the same.

**33. Page 21, table 3. This table includes the same information as shown in figure 9 I think but expanded for more thresholds. Could this be moved to the supplementary material?**

Yes, this is a good suggestion and we have moved this table to the supplementary material section of the paper.

**34. Page 22, figure 10. Why is the SO2 mass lost too fast in the VolRes1.5 simulation? Why is the grey arrow in panel (b) the same as in (a)? Shouldn't it have smaller vertical extent to indicate reduced amplitude spread? Why does the arrow in panel (d) and (d) extend beyond the point with the most negative structure value? Is it necessary to show the individual TROPOMI overpasses? Perhaps including the daily averaged squares only would be easier to follow the evolution in the SAL scores?**

We agree with the reviewer that the arrows might be confusing at the moment. They are to indicate the general direction into which the daily trends are moving. We have updated figure 10 to better represent the differences between the panels in the figure. We keep the individual points on the map, as we found that this shows essential information about the spread around these daily average values.

**35. Page 22, line 471. How do you know that the high VCDs are related to small-scale eddies?**

In the TROPOMI data these are shown as small-scale features, most of them are eddies. However, we agree with the reviewer that we cannot claim all the high VCDs are related to small-scale eddies and we have removed this statement from the manuscript.

**36. Page 23, line 479. The authors state that the S-values are 'relatively close to 0'.**
**Relative compared to what?**

We agree that the term 'Relatively' is confusing and should not have been used here. We have removed it from the manuscript.

**37. Page 23, line 480. Does the Location value have units or is it non-dimensional?**

It is non dimensional. It is the fraction of the domain width, where 1 means that the distance between the two features is the total domain width.

**38. Page 23, line 483. '4-5 days after the start of the eruption'. Please refer to the dates**
**used in the figure if possible.**

We have updated the text (L.518, L.525 and L.540) to refer to the corresponding dates in the figure.

**39. Page 23, line 490. What happens after 5 days?**

What happens after the 5 days (26 June) is addressed in the following sentence. We have now added the date (see previous point), which we think now clarifies the point.

**40. Page 23, line 493. What is the StratProfile simulation event 'better' than?**

Better than the VolRes1.5 and the VolRes2.0 simulations. We have changed to wording accordingly (L.529).

**41. Page 23, line 499. What is the motivation for reducing the K_meso value by 75%? Is there**
**evidence in the literature that this is appropriate or are you simply using K_meso as a tuning**
**parameter?**

We are indeed using it as a tuning parameter. In the literature there is a large range available for realistic diffusion parameters, which are also NWP wind field dependent. According to a study by Harvey et al. (2018) the possible values for sigma_meso are between 0.27 and 1.74 ms-1 (where in our reduced K_meso experiment we use sigma_meso 0.4 ms-1). However, there are no best values presented in this study and as pointed out by the other reviewer (see point 49 of their review), this study was applied to volcanic ash that might result in different values for the parameter.

Another study by Waugh et al. [JGR, 1997], defines an effective horizontal diffusivity of 10^3 m^2s-1 on the basis of observed mixing of a filament, identified in SPADE data. This value is close to our reduced K_meso estimate (1600 m^2s-1).

As we have limited literature available that gives any indication on the best values for K_meso, we have stepwise reduced the parameter and found the best results for the 75% reduction, which is the value presented in the paper. We have updated our manuscript to better explain our parameter choices in section 2.3.1 (L.206-221).

*42. Page 23, line 504. Why is the S score independent of the diffusion parameter? Is this because the size of the plume becomes greater than the size of the mesoscale eddies that K_meso is representing so the synoptic scale uncertainty dominates?*

Yes, the reviewer is correct. The uncertainties in the large-scale synoptic scale winds are dominating the signal on timescales over 5 days. As a result, the small-scale diffusion becomes irrelevant and all simulations behave equally well in terms of the S score. We have changed to text to represent this argument more clearly (L.541).

*43. Page 24, figure 11. Here the AAI is used to indicate high concentrations of ash, thereby affecting the TROPOMI SO2 retrievals. This interference is referred to at several points earlier in the paper but until this figure I don't think any evidence was shown to support the statement that ash was potentially contaminating the retrieval. Perhaps this evidence could be included earlier in the paper?*

We agree with the reviewer that evidence should be included when we make our earlier statements in the manuscript. We have reported the values for the AAI in section 2.1 for the first two days (L.152-153), which indicate the potential contamination of the $SO_2$ retrievals during the start of the eruption and refer to these values when making the statements later in the manuscript.

*44. Page 24, line 518. Why does the VolRes1.5 simulation loose SO2 mass at a much faster rate than TROPOMI?*

This is most likely related to the altitude of the volcanic $SO_2$ cloud in the VolRes1.5 simulation. In the VolRes1.5 simulation, $SO_2$ mass emitted into the troposphere came into contact with a cyclone and is for a large part removed (approximately 90%) through the wet deposition parameterisation (also indicated by the dashed line at the bottom of the figure 11a). When we run a simulation where we do not include this process, the mass reduction during the first week of the simulation is strongly reduced and the $SO_2$ mass burden for the VolRes1.5 simulation in figure 11a remains above the TROPOMI estimate.

We have updated the manuscript to clarify this point (L.580-599).

*45. Page 28, line 641. Do you change K_meso everywhere or just in the stratosphere?*

We change it everywhere in the free atmosphere (i.e., above the BL). We have included a line to clarify this point. (L.707)

*46. Page 28, line 644. Why do you think that a 'precise value for the diffusion parameters' exists? Turbulence is typically patchy, suggesting that a constant value is unsuitable.*

Yes, this is very well possible. But the current limitation in the model is that the diffusion is parameterised by a single value across the globe. Maybe this is not realistic and might need to be changed to a location dependent parameter in future versions of the model, but this is currently not a possibility for us to explore. We have added a short discussion of this point to the manuscript (L.699-713)

**47. Page 29, lines 681-684. This appears to be a repetition of the results already stated in the results section, not a conclusion.**

We agree with the reviewer that this part of the conclusions feels repetitive and have rewritten it to better represent the outcomes of our study.

**48. Page 30, line 695. It is surprising that no reference to the extensive literature on stratospheric dispersion is included here.**

We have now added some references to the literature on stratospheric dispersion (L.758-759).